# Mutant p53 regulates a distinct gene set by a mode of genome occupancy that is shared with wild type

Ramy Rahmé [ID] [1,2,3], Lois Resnick-Silverman [ID] [1], Vincent Anguiano [1,4], Moray J Campbell [ID] [5], Pierre Fenaux [2,6] & James J Manfredi [ID] [1,4 ✉]

## Abstract

To directly examine the interplay between mutant p53 or Mdm2 and wild type p53 in gene occupancy and expression, an integrated RNA-seq and ChIP-seq analysis was performed in vivo using isogenically matched mouse strains. Response to radiation was used as an endpoint to place findings in a biologically relevant context. Unexpectedly, mutant p53 and Mdm2 only inhibit a subset of wild type p53-mediated gene expression. In contrast to a dominant-negative or inhibitory role, the presence of either mutant p53 or Mdm2 actually enhances the occupancy of wild type p53 on many canonical targets. The C-terminal 19 amino acids of wild type p53 suppress the p53 response allowing for survival at sublethal doses of radiation. Further, the p53 mutant 172H is shown to occupy genes and regulate their expression via non-canonical means that are shared with wild type p53. This results in the heterozygous 172H/+ genotype having an expanded transcriptome compared to wild type p53 + /+.

Keywords p53; DNA Damage; Radiosensitivity; Gene Expression; Bone Marrow
Subject Categories Cancer; Chromatin, Transcription & Genomics; Genetics, Gene Therapy & Genetic Disease

## Introduction

The *TP53* gene (*Trp53* in mice) encodes the transcription factor p53 that binds DNA in a sequence-specific manner and activates transcription of target genes (Carvajal and Manfredi, 2013; Kruse and Gu, 2009; Vousden and Prives, 2009). Transcriptional regulation is closely associated with p53 tumor suppressor activities as cancer-associated missense mutants lose the ability to bind DNA (Brosh and Rotter, 2010; Freed-Pastor and Prives, 2012; Muller and Vousden, 2013). In the majority of cancer cases, missense mutations cause the substitution of a single amino acid in the

p53 protein (Olivier et al, 2010; Soussi and Wiman, 2007). Those mutants are stably expressed, but do not bind to wild type p53 consensus sites (Bargonetti et al, 1991; Bargonetti et al, 1993; el-Deiry et al, 1992; Funk et al, 1992; Kern et al, 1991b). The persistence of mutant p53 protein expression over mere loss suggests an inherent biological advantage for the mutants in human cancers (Olivier et al, 2010). This is in contrast to other tumor suppressor genes that undergo deletion through the course of tumor initiation or development such as *RB1* (Lan et al, 2020), *PTEN* (Yehia et al, 2020), and *BRCA1* or *BRCA2* (Stoppa-Lyonnet, 2016).

It has thus been proposed that these mutant p53 proteins are not merely loss-of-function, but rather have, in addition, acquired oncogenic activities that can contribute to tumorigenesis (Lang et al, 2004; Morton et al, 2010; Olive et al, 2004). A growing body of evidence supports the idea that the oncogenicity of mutant p53 involves affecting the transcription of a variety of genes involved in cell proliferation (Bossi et al, 2006; Bossi et al, 2008; Freed-Pastor and Prives, 2012; Haupt et al, 2009; Scian et al, 2004; Strano et al, 2002; Yan and Chen, 2009; Yan et al, 2008), resistance to apoptosis (Bossi et al, 2008; Lim et al, 2009), migration (Adorno et al, 2009; Weissmueller et al, 2014) and tissue invasion (Strano et al, 2002; Weissmueller et al, 2014). It is also likely that transcriptional regulation by mutant p53 depends on context. Tumor-specific alterations, cellular metabolism, chromatin landscape and availability of other specific transcription factors are likely to also affect mutant p53 activities (Adorno et al, 2009; Dell'Orso et al, 2011; Haupt et al, 2009; Kim and Deppert, 2003, 2004; Li et al, 2008; Rodriguez et al, 2012; Strano et al, 2007). Because p53 acts as a tetramer, these p53 mutants also have been characterized as having dominant-negative effects on any activity of the remaining wild type p53 in cancer cells (Aubrey et al, 2018; Gencel-Augusto and Lozano, 2020).

In contrast to cancer cells, mutant p53 activities are not well defined in nonmalignant cells. To examine this in more detail, the acute response to DNA damage was studied in murine bone marrow harboring the p53 missense mutation R172H (equivalent of the human hotspot mutant R175H found in cancer). This led to the unexpected finding that wild type and mutant p53 may share non-canonical means to regulate gene expression in response to

[1]Department of Oncological Sciences and Tisch Cancer Institute, Icahn School of Medicine at Mount Sinai, New York, NY 10029, USA. [2]Institut de Recherche Saint Louis (IRSL), INSERM U1131, Université de Paris, Paris, France. [3]Ecole Doctorale Hématologie–Oncogenèse–Biothérapies, Université de Paris, Paris, France. [4]The Graduate School of Biomedical Sciences, Icahn School of Medicine at Mount Sinai, New York, NY 10029, USA. [5]Cedars-Sinai Medical Center, Los Angeles, CA 90048, USA. [6]Service Hématologie Seniors, Hôpital Saint Louis, Assistance Publique-Hôpitaux de Paris (AP-HP), Université de Paris, Paris, France. ✉E-mail: james.manfredi@mssm.edu

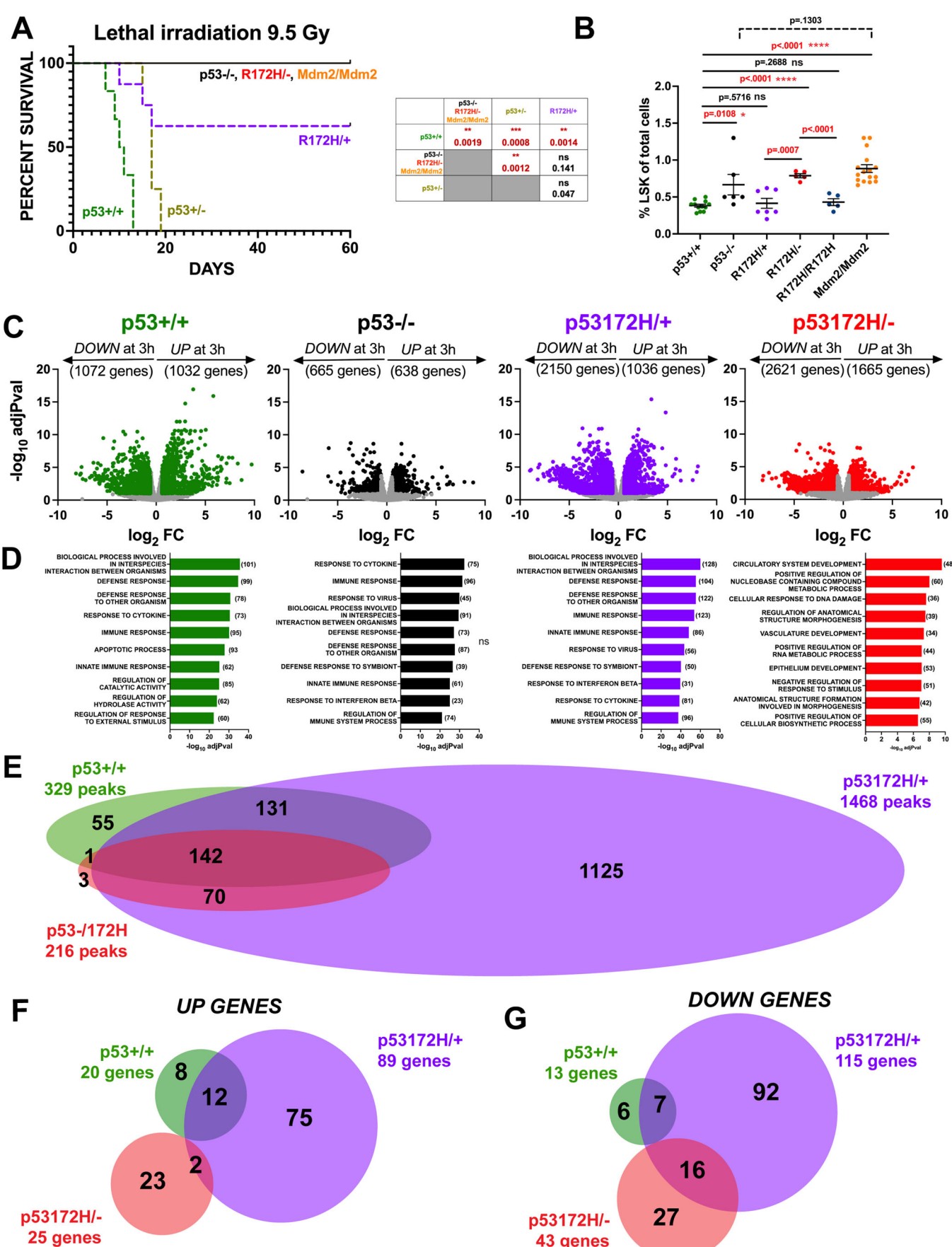

**Figure 1.  Sensitivity to a lethal dose of radiation is p53-dependent.**

(A) Kaplan–Meier survival curves of lethally irradiated mice: Trp53-WT ($n = 6$); Trp53 WT/null ($n = 6$); Trp53-null ($n = 5$); R172H/WT ($n = 8$) and R172H/null ($n = 5$). For surviving mice, monitoring stopped two months after radiation treatment. Log-rank tests were performed on survival plots. ****$P < 0.0001$; ***$P < 0.001$; **$P < 0.01$; *$P < 0.05$; ns, not significant. Significant $P$ values are shown in red. p53$+/+$ vs p53$-/-$ and R172H/$-$ and Mdm2/Mdm2: $P = 0.0019$**; p53$+/+$ vs p53$+/-$: $P = 0.0008$***; p53$+/+$ vs p53 R172H/$+$: $P = 0.0014$**; p53 $+/-$ vs p53$-/-$ and p53R172H/$-$ and Mdm2/Mdm2: $P = 0.0012$**; p53R172H/$+$ vs p53$-/-$ and p53R172H/$-$ and Mdm2/Mdm2: $P = 0.141$(ns); p53 $+/-$ vs p53R172H/$+$: $P = 0.047$(ns). (B) The frequency of Lin($-$) Sca1($+$) Kit($+$) cells (LSK) assessed by flow cytometry as a percentage of total BM cells in Trp53-WT ($n = 12$), Trp53-null ($n = 6$), R172H/WT ($n = 7$) and R172H/null ($n = 5$) mice. Data are represented as mean ± SE. $P$ values are calculated using one-way ANOVA with Dunnett's multiple comparisons test. ****$P < 0.0001$; ***$P < 0.001$; **$P < 0.01$; *$P < 0.05$; ns, not significant. Significant $P$ values are shown in red. p53$+/+$ vs p53$-/-$: $P = 0.0108$*; p53$+/+$ vs p53R172H/$+$: $P = 0.5716$(ns); p53$+/+$ vs p53 R172H/$-$: $P < 0.0001$****; p53$+/+$ vs p53R172H/R172H: $P = 0.2688$(ns); p53$+/+$ vs Mdm2/Mdm2: $P = {<}0.0001$****; p53$-/-$ and Mdm2/Mdm2: $P = 0.1303$(ns). (C) Eight-week-old mice of the indicated genotype were untreated or treated with 9.5 Gy of X-ray. After 3 h, RNA was extracted from bone marrow, and subjected to RNA-seq analyses. Volcano plots are shown as -$\log_{10}$ adjusted $P$ value (adjPval) versus $\log_2$ Fold change (FC). Data is derived from $N = 3$ mice. Differentially expressed genes upon radiation with an adjusted $P$ value (False Discovery Rate) <0.1 and an absolute fold change >1.5 are shown as colored dots. Statistical analysis for differential gene expression analysis was performed using DESeq2 (Love et al, 2014). (D) GSEA analysis was performed for the significant differentially expressed up-regulated genes and the top Gene Ontology Biological Processes terms are shown with corresponding -$\log_{10}$ adjPval. Statistical analysis for Gene Set Enrichment Analysis (GSEA) was performed using web tools provided by The University of California, San Diego and the Broad Institute (https://www.gsea-msigdb.org/gsea) (Mootha et al, 2003; Subramanian et al, 2005). (E) 8 week-old mice of the indicated genotype were treated with 9.5 Gy of X-ray. After 3 h, bone marrows were subjected to cross-linking and ChIP-seq analyses. A Venn diagram is shown for genotype-specific and overlapping ChIP-seq peaks with an adjusted $P$ value <0.2. Statistical analysis for ChIP peals was performed using ChIPpeakAnno (Zhu et al, 2010). (F, G) Venn diagrams for upregulated (F) or downregulated (G) genes are shown that have associated ChIP-seq peaks that are within 10 kb of the gene TSS. Data information: In (B), data are presented as mean ± SE. *$P < 0.05$ (one-way ANOVA). Source data are available online for this figure.

radiation, and this is associated with dominant-negative effects on only a small subset of wild type p53 target genes.

## Results

### Sensitivity to a lethal dose of radiation is p53-dependent

A p53-dependent model system was needed in which biological outcomes can be directly related to gene occupancy and transcriptional output. Further, the ideal model would allow for isogenic comparisons with minimal, if any, other genetic differences between wild type, null and mutant p53 conditions. While cell-based models might be suitable, in vivo studies in the mouse ensure that the various genotypes are as isogenic as possible. The goal is also to definitively establish gene occupancy by mutant p53. As the molecular basis for mutant p53 gene occupancy remains unclear, this experimental system was chosen to allow a direct comparison between wild type and mutant ChIP-seq with the only difference being genotype. Previous studies have shown that the response of mice to whole body radiation with the commonly used lethal dose of 9.5 Gy is strictly p53-dependent (Fei et al, 2002; Gudkov and Komarova, 2003; Kirsch et al, 2010; Lee and Bernstein, 1993; Lowe et al, 1993; Merritt et al, 1994). The focus on the 172H mutant is twofold. First, 172R in the mouse (175R in humans) is one of six residues which are hotspots for missense mutation in human cancers. Second, 172H together with 270R, another hotspot in human tumors (273R in humans), were the first to be modeled in the mouse and these genetically engineered strains are well characterized and widely available.

With this in mind, eight-week-old mice were treated with a combined lethal X-ray dose of 9.5 Gy. Both wild type ($+/+$) and heterozygous null ($+/-$) mice die within three weeks after treatment. Mice that lack p53 expression ($-/-$), express a missense mutant p53 in the absence of wild type (172H/$-$), or express high levels of Mdm2 (Mdm2/Mdm2) do not succumb to this lethal dose and survive for longer than two months. The heterozygous mutant mice (172H/$+$) show an intermediate response (Fig. 1A).

Reconstitution of such lethally irradiated mice with untreated $+/+$ bone marrow cells completely rescues the outcome, indicating that hematopoietic failure is central to the wild type response. Differences in LSK cells (i.e. the hematopoietic stem and progenitor cells in mice) were then compared among the studied genotypes. For instance, given the percentage of LSK cells was similar in $+/+$ and 172H/$+$ bone marrows, meaning both genotypes have the same recovery potential after radiation, quantitative differences were not sufficient to explain the different outcomes between these two genotypes (Fig. 1B).

First, transcriptomic changes induced by radiation in $+/+$, $-/-$, 172H/$+$ and R172H/$-$ bone marrows were studied. This enabled direct assessment of the relationship between DNA binding and transcriptional regulation. Up- and downregulated genes with an adjusted $P$ value < 0.1 and an absolute fold-change >1.5 were considered as differentially expressed. Transcriptional changes observed at three hours after radiation are shown in Fig. 1C. Gene Set Enrichment Analysis (GSEA) of biological processes associated with differentially expressed genes was conducted. In irradiated $+/+$ bone marrows, enriched gene sets were consistent with canonical p53 responses to radiation (Fig. 1D). Interestingly, Gene Ontology terms enriched in 172H/$-$ differed from those found in $-/-$, suggesting the existence of specific mutant p53-dependent pathways in response to high dose of radiation (Fig. 1D). Immunoblotting to detect p53 protein was unsuccessful (Appendix Fig. S1) making it difficult to rule out expression level as the basis for these differences. To gain insight into relative protein expression levels of the wild-type and mutant proteins, spleen and thymus from mice irradiated under the same conditions as the bone marrow studies were examined. Immunoblotting for protein (Fig. EV1) was performed. Wild-type and mutant p53 protein expression was detectable and is similar in both tissues, both before and after treatment (Fig. EV1A). Taken together, these new data provide two additional insights. First, they confirm that effects of mutant p53 are not due to differences in protein expression. Rather, observed effects are likely due to differences in p53 activity. Second, they show that mutant p53 protein is induced to comparable levels as wild-type regardless of genotype of tissues (m/$+$, m/$-$).

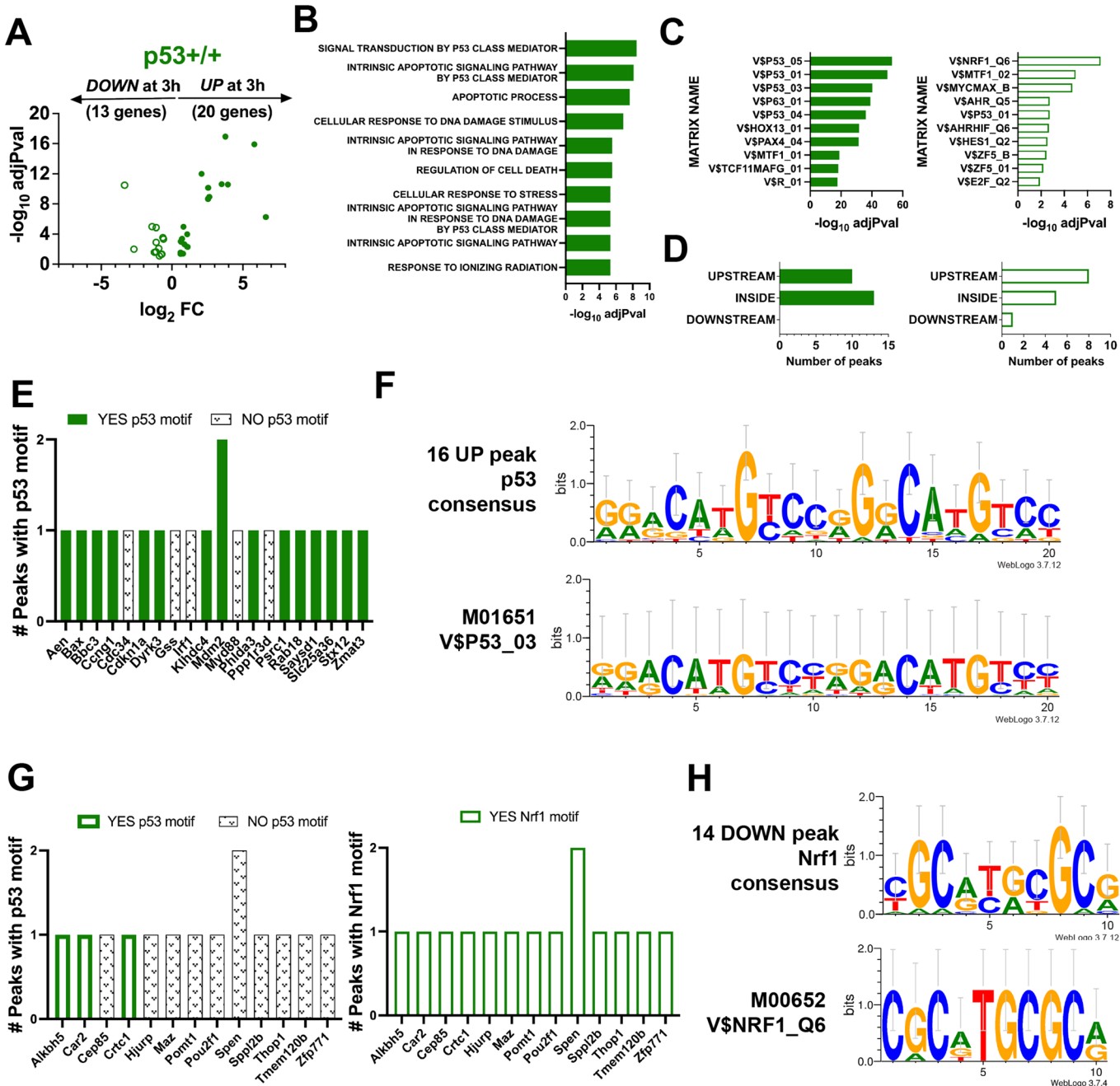

ChIP-seq analyses were then performed. In all, 329 peaks were assigned to genes in +/+, 1468 in R172H/+ and 216 in R172H/−. Figure 1E shows the Venn diagram for the ChIP-seq results with peak overlaps between the three genotypes. The ChIP-seq and RNA-seq results in irradiated bone marrows were then integrated. For this analysis, targets, the cistromic genes, were defined as up- or downregulated differentially expressed genes with a ChIP peak that is located within 10 kb of the transcription start site (TSS) of the gene. Venn diagrams for upregulated (Fig. 1F) or downregulated (Fig. 1G) genes are shown. No up and down targets were shared between the three genotypes. The comparison between +/+ and 172H/− revealed the existence of exclusive cistromic genes for

mutant p53. In all, these results uncovered the unexpected finding that mutant p53 is likely to directly and specifically regulate gene expression in response to radiation.

## Wild type p53 occupies and activates genes in response to X-radiation

To ensure that the experimental model behaves as expected, the gene occupancy and expression in the wild type bone marrow was examined first. A small number of bona fide targets can be identified by integrating RNA-seq and ChIP-seq data sets (Fig. 2A). The resulting targets are low in number (20 upregulated and 13

**Figure 2.   Wild type p53 occupies and activates genes in response to X-radiation.**

(A) A Volcano plot is shown for upregulated or downregulated genes in p53 + /+ bone marrow is shown that have associated ChIP-seq peaks that are within 10 kb of the gene TSS. Data is derived from N = 3 mice. Statistical analysis for differential gene expression analysis was performed using DESeq2 (Love et al, 2014). (B) GSEA analysis was performed for the significant differentially expressed up-regulated genes and the top Gene Ontology Biological Processes terms are shown with corresponding -log$_{10}$ adjPval. Statistical analysis for Gene Set Enrichment Analysis (GSEA) was performed using web tools provided by The University of California, San Diego and the Broad Institute (https://www.gsea-msigdb.org/gsea) (Mootha et al, 2003; Subramanian et al, 2005). (C) The genomic sequences underneath the peaks detected by ChIP-seq for either the upregulated or downregulated genes were subjected to TRAP analysis. The top matrices that were detected with the corresponding adjPvalues are shown. Statistical analysis for motif analysis was done by TRAP (Transcription Factor Affinity Prediction) using web tools provided by the Max Planck Institute for Molecular Genetics (http://trap.molgen.mpg.de) (Thomas-Chollier et al, 2011). (D) The location of the ChIP peaks associated with each of the differentially expressed genes is shown. (E) For the 20 upregulated genes, the number of peaks within 10 kb of the transcription start site that are detected by ChIP-seq for each gene is shown. The solid bars show peaks which contain a consensus p53 motif, while the hatched bars show peaks that do not. (F) The p53 motifs from the 16 peaks with a canonical p53 motif were subjected to WebLogo3 analysis and the resulting consensus is shown. For comparison, the M01651 matrix is also shown. (G) For the 13 downregulated genes, the number of peaks within 10 kb of the transcription start site that are detected by ChIP-seq for each gene is shown. On the left, the colored bars show peaks which contain a consensus p53 motif, while the hatched bars show peaks that do not. On the right, the colored bars show peaks which contain a consensus Nrf1 motif. (H) The Nrf1 motifs from the 14 peaks with a canonical Nrf1 motif were subjected to WebLogo3 analysis and the resulting consensus is shown. For comparison, the M00652 matrix is also shown. Source data are available online for this figure.

downregulated cistromic genes), but are likely to represent biologically relevant transcriptional changes since RNA-seq cut-offs are combined with a more rigorous criterion of a corresponding ChIP peak within 10 kb of the transcriptional start site (TSS). This is likely excluding genes that are also important, but nevertheless increases the likelihood that the genes being studied contribute to observed phenotypes. GSEA of the 20 up and bound genes is consistent with a wild type p53 response (Fig. 2B). TRAP (TRanscription factor Affinity Prediction) (Thomas-Chollier et al, 2011) analysis for these upregulated targets likewise is consistent with p53 occupancy at genes with canonical response elements (Fig. 2C). The location of the ChIP peaks (Fig. 2D) and the number of peaks for each of the 20 upregulated genes that contain a canonical p53 motif (Fig. 2E) are shown. Using the latter, a consensus can be generated which matches closely to published matrices for *TP53* (Fig. 2F). Concerning the 13 downregulated genes (Fig. 2A), TRAP analysis surprisingly did not show strong matches to p53 matrices, but rather to that of Nrf1 (Fig. 2C). Indeed, few peaks associated with the downregulated genes had a strong match to the canonical p53 consensus, but rather all of the peaks had canonical Nrf1 motifs (Fig. 2G). A consensus of these sequences in the 13 downregulated genes matches closely to a published matrix for *NRF1* (Fig. 2H).

## Mutant p53 and Mdm2 only inhibit a subset of wild type p53-mediated gene expression in a manner independent of gene occupancy

The transcriptional response for these 20 genes was then examined with the various genotypes that show full radioresistance: as expected, none of these genes are up in p53−/− or 172H/− bone marrows (Fig. 3A). Surprisingly, the majority of these genes are also up in both the 172H/+ and the Mdm2/Mdm2 mice (Fig. 3B) even though these mice show partially or complete resistance to the lethal dose of radiation (Fig. 1A). There are six genes that are up and bound in the p53 + /+ tissues, but not in these others (*Cdc34*, *Dyrk3*, *Gss*, *Klhdc4*, *Rab18*, and *Stx12*) (Fig. 3C), suggesting that these may be important for radiosensitivity or, at least, are part of a signature. The genes that are still upregulated by p53 in the presence of either the mutant protein or Mdm2 (Fig. 3B) include p53 canonical targets such as *Bax*, *Bbc3*, *Cdkn2a*, and *Mdm2* itself (Fig. 3C). Hence, neither Mdm2 overexpression nor mutant p53

expression were able to prevent transcriptional activation by p53 of these canonical targets at 3 h after radiation.

RT-qPCR analysis of bone marrow confirms findings with the various genotypes (p53 + /+; p53m/+; p53 m/−; and Mdm2 Tg/Tg) RNA-seq for three targets (*Cdkn1a*, *Bbc3*, and *Ccng1*) (Fig. EV2A–F). Spleen and thymus were also examined and show, with either the wild-type or the heterozygous m/+ genotypes, robust activation of mRNA after irradiation (Fig. EV2A–C) and this is reflected in protein increases for one target, p21 (*Cdkn1a*) (Fig. EV1A). As expected, tissues from the m/− mice show no mRNA upregulation (Fig. EV2A–C), nor increases in p21 protein (Fig. EV1A). In the Mdm2/Mdm2 tissues, upregulation of p21 protein is impaired, however modest increases in Puma are retained (Fig. EV1C). To examine effects on p53 protein expression, immunoblotting was performed in spleen and thymus, since it is difficult to detect p53 in bone marrow. Overexpression of Mdm2 in the transgenic mouse model appears to modestly affect p53 levels in spleen and thymus under these same conditions (Fig. EV1C).

Intriguingly, ChIP peaks were stronger in 172H/+ but weaker in Mdm2/Mdm2 (Fig. 3E), as illustrated for *Zmat3* (Fig. 3F). In addition, gene occupancy was not always associated with gene expression as showed for the "p53 only genes" *Cdc34* and *Stx12* (Fig. 3F). In parallel, expression of the thirteen downregulated cistromic genes identified in +/+ bone marrow after radiation was also assessed (Appendix Fig. S2A–D). ChIP peaks were likewise stronger in 172H/+ (Appendix Fig. S2E) and gene occupancy was not necessarily associated with downregulation (e.g. *Alkbh5*) (Appendix Fig. S2F). Unexpectedly, while Nrf1 peaks were observed in only 6 up genes (Fig. 3D), almost all down genes showed a known Nrf1 peak (Appendix Fig. S2D). These findings do not support the idea that mutant p53 inevitably prevents wild type p53 from binding to DNA consensus sites in response to radiation in vivo.

## The C-terminal 19 amino acids suppress the p53 response at sublethal doses of radiation

To gain further insight into the mechanisms of radiosensitivity in +/+ bone marrow, we considered the role of the p53 C-terminal domain (CTD). It had been previously shown that deletion of the p53 CTD in mice (the terminal 24 amino acids, referred to as Δ24, Fig. 4G) (Hamard et al, 2013) causes lethality by 2 weeks of age that

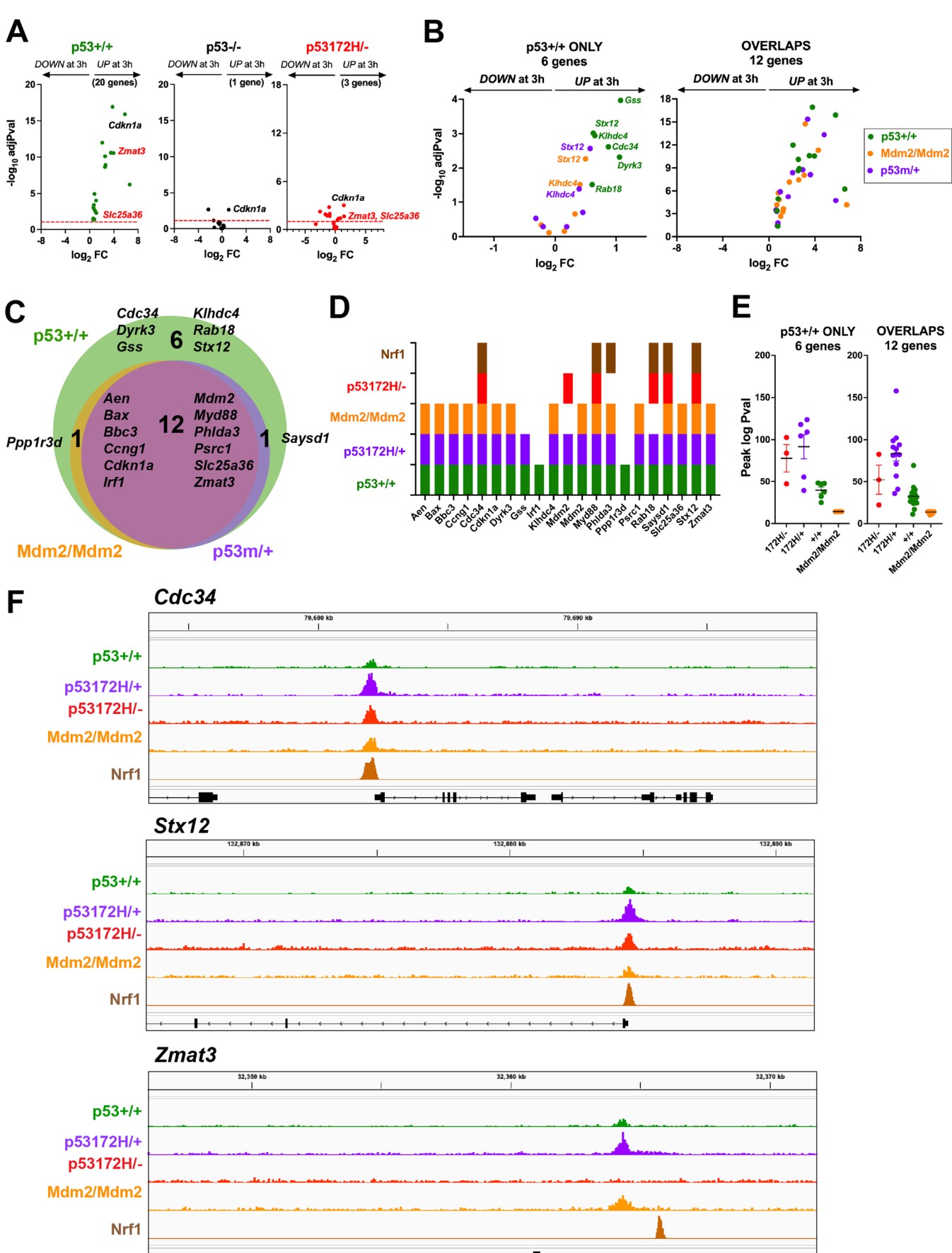

◀ **Figure 3.  Mutant p53 and Mdm2 only inhibit a subset of wild type p53-mediated gene expression.**

(A) The expression of the 20 upregulated genes in p53 + /+ bone marrow was examined in either p53−/− or p53172H/− mice. Corresponding Volcano plots are shown. Data is derived from *N* = 3 mice. Statistical analysis for differential gene expression analysis was performed using DESeq2 (Love et al, 2014). (B) The expression of the 20 upregulated genes in p53 + /+ bone marrow was examined in either p53172H/+ or the Mdm2/Mdm2 mice. Corresponding Volcano plats are shown. Two classes of genes are identified: those which are only upregulated in p53 + /+ mice (left) and those which also differentially regulated in either the p53172H/+ or the Mdm2/Mdm2 mice (right). Data is derived from *N* = 3 mice. Statistical analysis for differential gene expression analysis was performed using DESeq2 (Love et al, 2014). (C) A Venn diagram shows the two classes of genes identified in (B). (D) For each of the 20 upregulated genes in p53 + /+ bone marrow, the presence of a corresponding ChIP peak for each indicated genotype is shown. Nrf1 occupancy is taken from an ENCODE data set for MEL cells (Accession: ENCSR135SWH). (E) A plot of the corresponding adjPval for the ChIP peaks associated with each gene set from the ChIP-seq analysis for each genotype is shown. Statistical analysis for ChIP peaks was performed using ChIPpeakAnno (Zhu et al, 2010). Error bars are SEM. (F) ChIP profiles of three such genes visualized in the IGV Browser are shown. Source data are available online for this figure.

is primarily due to hematopoietic failure. Hence, these mice cannot be used to perform radiosensitivity studies such as these. Two additional mouse strains were engineered using CRISPR/Cas9 technology to have deletion of either 19 (Δ19) or 10 (Δ10) residues from the C-terminus (Fig. 4G). These mice are fully healthy, do not show any blood cell count abnormalities, and thus can be used to test radiosensitivity.

Although the dose of 9.5 Gy is lethal in wild type p53 mice, 6 Gy is not (Fig. 4A). However, both Δ19 and Δ10 strains show enhanced sensitivity to sublethal radiation with 6 Gy (Fig. 4A). Necropsy studies (wild type mice were euthanized at day 10 after radiation) showed all the groups had similarly depleted bone marrows (Appendix Fig. S3B). Conversely, the mutant groups had more severe gastric and myocardial injuries that could also have contributed to death (Appendix Fig. S3B). Importantly, the radiosensitivity of Δ19 mice was largely rescued with as little as 50,000 wild type bone marrow cells, but not with the corresponding number of Δ19 cells (Fig. 4B), indicating that bone marrow failure is essentially responsible for lethality. It should be noted that it is critical to titrate the appropriate number of cells to reveal such differences, which otherwise might not be initially apparent (Fig. 4B). This observation was further explained by differences in LSK cells with Δ19 mice showing the lowest percentage compared to +/+ (*P* < 0.001) (Fig. 4F).

The transcriptomic signature associated with lethality was then examined using RNA-seq. The transcriptional response to 6 Gy of radiation was more substantial in the Δ19 mice as compared to wild type with almost four times as many genes being upregulated at 6 Gy with genetically altered mice (Fig. 4C). Examination of genes that are differentially expressed between wild type and Δ19 bone marrows after 6 Gy of radiation corresponded to distinct gene ontology terms, further emphasizing that deletion of the terminal 19 amino acids has likelyu biological consequences (Fig. 4E). Transcriptomic changes in +/+ bone marrows after sublethal (6 Gy) (Fig. 4C) and lethal (9.5 Gy) (Fig. 1C) dose of radiation, as well as in Δ19 after exposure to a lethal dose of 6 Gy (Fig. 4C) were then compared. (Fig. 4D). 591 genes were identified as being upregulated by both 6 Gy and 9.5 Gy in wild-type mice. Of these 591 genes, only 391 were also upregulated in the Δ19 mice at 6 Gy as well (Fig. 4D). Interestingly, the majority of the upregulated genes (2613 out of 3313) in the Δ19 mice were unique to this genotype.

Unlike full-length wild-type p53, protein expression of both the Δ10 and Δ19 engineered forms of wild-type p53 can be detected. Although there is some variation from mouse to mouse, it appears that both forms are upregulated in response to radiation and their

levels of expression are substantially higher than that of wild-type p53 (Appendix Fig. S3). This altered expression is a likely explanation for the radiation-related phenotypes of the mice expressing the truncated proteins (Fig. 4A,B,F). Nevertheless, the RNA-seq studies identify genes that are selectively upregulated in p53 + /+ bone marrow (Fig. 4C,D). This suggests that there may also be intrinsic differences inp53 activity between the wild-type and Δ19 proteins.

## Mutant 172H p53 occupies genes and regulates their expression after X-radiation via non-canonical means for gene occupancy that is shared by wild type and mutant p53

A corresponding integrated RNA-seq and ChIP-seq analysis was performed on the bone marrow of the 172H/− mice after radiation treatment (Fig. 5A). Twenty-four genes were occupied and upregulated in the 172H/− bone marrow with a corresponding 30 genes being occupied and downregulated (Fig. 5A). GSEA for the 24 up genes shows enrichment in Biological Process terms that include cell cycle and negative regulation of metabolism (Fig. 5B) distinct from what is seen with 172H/+ (Fig. 6B). The genomic sequences underneath the peaks detected by ChIP-seq for the upregulated genes were subjected to TRAP analysis. The top matrices that were detected were not of p53, but rather others, with a main hit being Nrf1 (Fig. 5C). Identification of putative Nrf1 motifs in these 24 genes showed that 22 of them had a detectable motif (Fig. 5D) that matched well with the reported canonical Nrf1 matrix (V$NRF1_Q6, Fig. 5E). Overlaying the p53 ChIP-seq and Nrf1 ChIP-seq data sets show that 22 out of the 24 genes had matching peaks. It should be noted that the two genes lacking Nrf1 peaks (*Zfp106* and *Zfp65*, Fig. 5F) are distinct from the ones that computationally lacked the motif (*Ift80* and *Mga*, Fig. 5D). Given the overlap in peaks seen previously with the *Zbtb41* gene (Fig. 6F), it was then determined whether any of these 24 genes also had peaks in the ChIP-seq data sets of the other genotypes. All 24 genes had matching peaks with both the 172H/+ and 172H/− genotypes, as well as 15 genes also showing peaks with the +/+ bone marrow (Fig. 5G). Only one of these 15 genes (*Birc2*) was also regulated in the +/+ bone marrow (Fig. 5H). For the 172H/+ genotype, just four additional genes were upregulated (*Nr1d2*, *Zbtbd41*, *Msh3*, and *Rps6kb1*, Fig. 5H). The ChIP-seq track for a gene that is only upregulated in the 172H/− bone marrow, *Xpo5* (Fig. 5I,J), confirms occupancy with all genotypes, as well as a corresponding Nrf1 ChIP peak (Fig. 5I). Taken together, this indicates that mutant p53, in the absence of any wild type p53 expression,

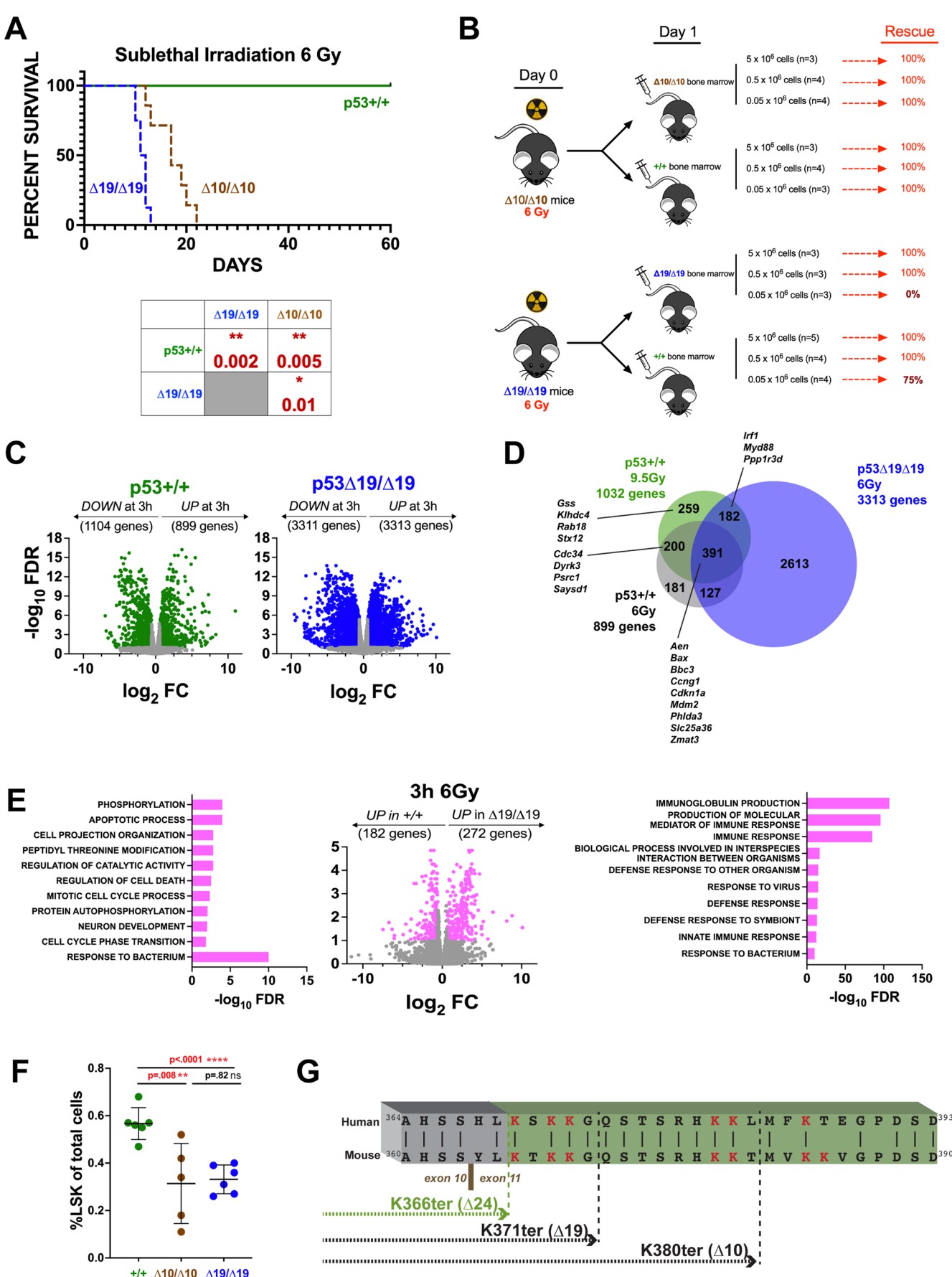

◄

**Figure 4. The C-terminal 19 amino acids suppress the p53 response at sublethal doses of radiation.**

(A) 8 week C57BL/6 mice with indicated genotypes received total body X-radiation of a single sublethal dose of 6 Gy. Log-rank tests were performed on survival plots. The significance level was adjusted using the Bonferroni method. *$P < 0.008$ (Bonferroni-corrected threshold) ($N = 8$). ****$P < 0.0001$; ***$P < 0.001$; **$P < 0.01$; *$P < 0.05$; ns, not significant. Significant $P$ values are shown in red. p53 + /+ vs p53Δ10/Δ10: $P = 0.005$**; p53 + /+ vs p53Δ19/Δ19: $P = 0.002$**; p53Δ10/Δ10 vs p53Δ19/Δ19: $P = 0.01$*. (B) Mice were X-radiated with a single dose of 6 Gy and then 3 h later bone marrow from the indicated genotypes were injected in the tail vein. Mice were monitored for survival on a daily basis ($N = 4$). (C) 8 week-old mice of the indicated genotype were untreated or treated with 6 Gy of X-ray. After 3 h, RNA was extracted from bone marrow, and subjected to RNA-seq analyses. Volcano plots are shown as $-\log_{10}$ adjusted $P$ value (adjPval) versus $\log_2$ Fold change (FC). Data is derived from $N = 3$ mice. Differentially expressed genes upon radiation with an adjusted $P$ value (False Discovery Rate) <0.1 and an absolute fold change >1.5 are shown as colored dots. (D) A Venn diagram shows overlapping significant genes for the gene sets shown in (C) and the genes shown in Fig. 1C for the $+/+$ genotype. (E). Differentially expressed genes between the two genotypes at 3 h after 6 Gy X-raidation are shown as a Volcano lot. Genes with an adjusted $P$ value (False Discovery Rate) <0.1 and an absolute fold change >1.5 are shown as pink dots. Data is derived from $N = 3$ mice. GSEA analysis was performed and the top Gene Ontology Biological Processes terms are shown with corresponding -$\log_{10}$ adjPval for the significant differentially expressed upregulated (right) or downregulated (left) genes. Statistical analysis for differential gene expression analysis was performed using DESeq2 (Love et al, 2014). Statistical analysis for Gene Set Enrichment Analysis (GSEA) was performed using web tools provided by The University of California, San Diego and the Broad Institute (https://www.gsea-msigdb.org/gsea) (Mootha et al, 2003; Subramanian et al, 2005). (F) The frequency of Lin(−) Sca1(+) Kit(+) cells (LSK) assessed by flow cytometry as a percentage of total BM cells in indicated genotypes. For $+/+$ and 10/Δ10, $N = 5$ mice; for Δ19/Δ19, $N = 6$ mice. Data are represented as mean ± SE. $P$ values are calculated using one-way ANOVA with Dunnett's multiple comparisons test; ****$P < 0.0001$; ***$P < 0.001$; **$P < 0.01$; *$P < 0.05$; ns, not significant. Significant $P$ values are shown in red. p53 + /+ vs p53Δ10/Δ10: $P = 0.008$**; p53 + /+ vs p53Δ19/Δ19: $P < 0.0001$****; p53Δ10/Δ10 vs p53Δ19/Δ19: $P = 0.82$(ns). (G) Schematic of p53 C-terminus and the nature of the Δ10 and Δ19 alleles. Data information: In (F), data are presented as mean ± SE. *$P < 0.05$ (one-way ANOVA). Source data are available online for this figure.

occupies and upregulates a subset of genes after radiation, almost all of which are occupied by mutant p53 in a region that is also occupied by Nrf1 and contains a canonical Nrf1 motif, but no consensus p53 motif. Intriguingly, many of these genes are also occupied by wild type p53 but fail to be transcriptionally regulated (Fig. 5G,H).

It should be noted that the relevant transcription factor here is Nuclear Respiratory Factor 1 which is encoded by the mouse *Nrf1* gene. There is some confusion in the published literature as Nuclear Factor, Erythroid 2-Like 1, encoded by the mouse *Nfe2l1* gene, is often referred to by the common name Nrf1. Nfe2l1 and its related Nfe2l2 are distinct transcription factors with quite different consensus binding sites from the bona fide Nrf1 (Appendix Fig. S4). It is thus unfortunate that they can be found in published studies as being called Nrf1 and Nrf2.

The enrichment of the Nrf1 motif under these ChIP peaks raises the question of the specificity of the CM5 antibody. To address this issue, three experiments were performed. First, an immunoblot using CM5 that shows the full gel without cropping was performed to show that the only detectable band is that corresponding to the molecular weight of p53 (both wild-type and mutant) (Fig. EV1B). Second, Use of tissues from p53-null mice shows that this band is no longer seen (Fig. EV1C). Third, extracts of mouse cells expressing mutant p53 were immunoprecipitated with CM5 and then these were subsequently immunoblotted with CM5. The only detectable band corresponds to the size of p53 (Fig. EV3A,C). Further, use of increasing amounts of extract, show a corresponding increase in the intensity of this band (Fig. EV3B,C). Taken together, these additional data argue in favor of CM5 being specific for full-length p53.

An additional concern is that CM5 may cross-react with Nrf1. To address this, two additional approaches were taken. First, there are no detectable bands with the migration size of Nrf1 (67kd) in immunoblots of tissues of various genotypes that have been probed with CM5 (Fig. EV1B). Second, an antibody to Nrf1 was used. Immunoprecipitation of extracts of mouse cells expressing mutant p53 with anti-Nrf1 and then subsequently probing with Nrf1 results in detection of a band of the appropriate molecular weight. Increasing amount of extract shows a corresponding increase in the

intensity of this band, arguing that the antibody is detected Nrf1 (Fig. EV3B). When such Nrf1 immunoprecipitates are probed with CM5, no bands of the size corresponding to Nrf1 are detected (Fig. EV3A–C). This confirms that it is unlikely that CM5 is cross-reacting with Nrf1.

There appears to be some similarities between the p53 and Nrf1 consensus motifs (Appendix Fig. S4). It is possible that multiple Nrf1 motifs in close proximity may be creating a p53 binding site. This was tested (Figs. EV4 and 5). For this, all ChIP peaks were examined, regardless of association with regulated genes (Fig. 1E). The genomic occupancies of p53 + /+ and p53-/172H were compared. A Venn diagram was generated to show specific and overlapping peaks for each genotype (Fig. EV4A). TRAP analysis for each set of peaks revealed that the +/+ only peaks were enriched for p53 motifs, whereas the shared and m/− peaks were high for Nrf1 motifs (Fig. EV4A,B). Of the 186 +/+ peaks, 84 contained a match to the p53 consensus (Figure EV4C). For the shared (143 peaks) and the m/− only (73 peaks), the majority had sequences matching an NRF1 consensus (139, and 72, respectively) (Fig. EV4C). We then examined the surrounding 50 bp as requested by the reviewer. There did not appear to be any further detectable consensus sequences beyond the 10 bp NRF1 motif (Fig. EV4D). These data further support the central finding of this study, that wild-type and mutant p53 share non-canonical means for genomic occupancy.

To validate the ability of such analyses to detect bona fide p53 motifs, peaks that contain p53 motifs that are shared between m/+ and +/+ (78 out of 131 peaks) versus m/+ alone (232 out of 1125 peaks) were examined (Fig. EV5). The resulting consensus sequences match well to known p53 motifs (compare the 84 peak consensus in Fig. EV4C with that of both sets of peaks in Fig. EV5C). This contrasts with the consensus matches to the Nrf1 motif (Fig. EV4C). Although most of the shared peaks has a p53 motif, intriguingly only a subset of the m/+ only peaks did. This suggest that perhaps the m/+ genotype results in yet another non-canonical mode for p53 genomic occupancy. These data also provide support for the idea that the m/+ genotype results in an expanded genomic occupancy as compared to that of the +/+ genotype (see below).

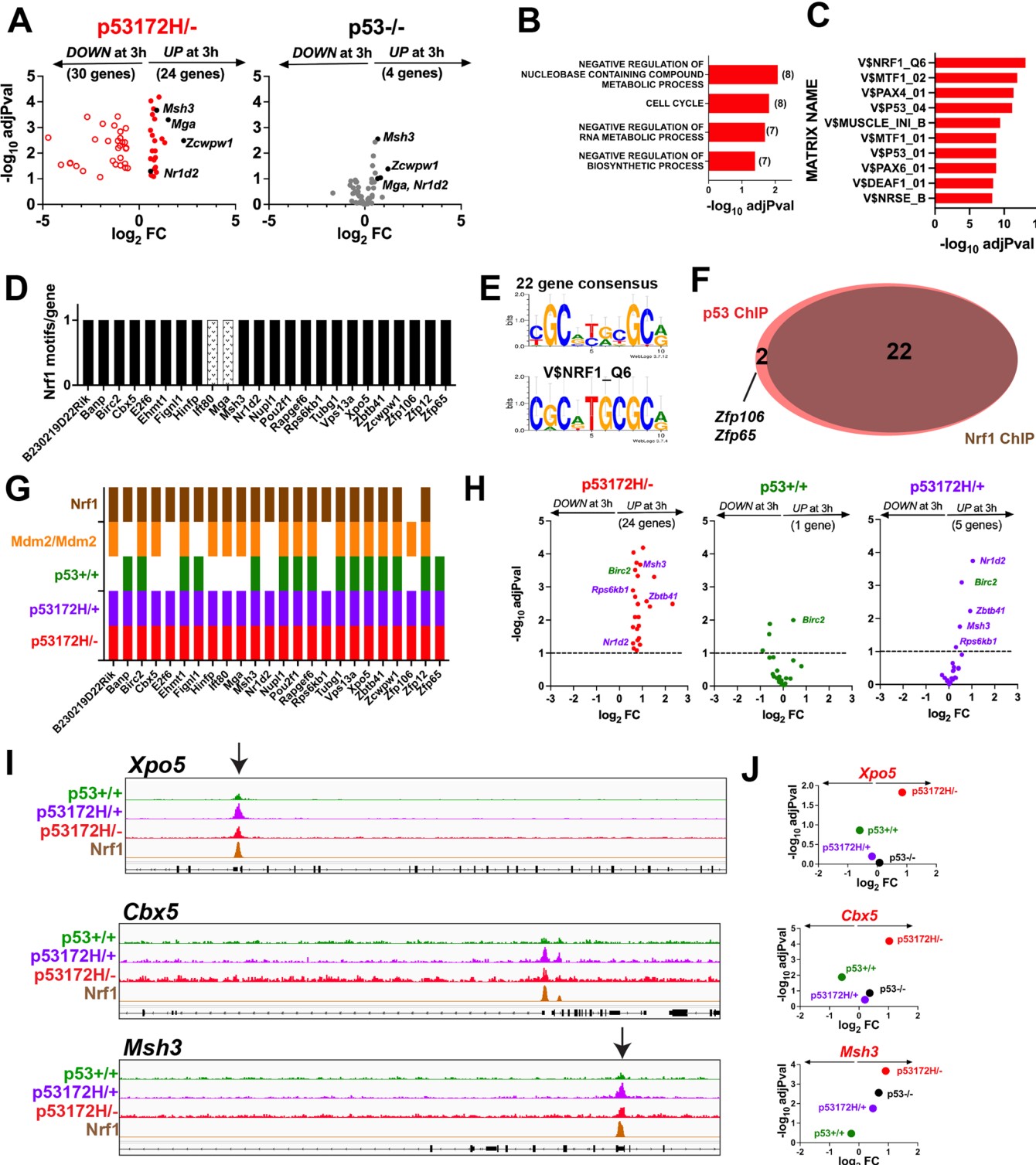

The possibility of a physical interaction between mutant p53 and Nrf1 would provide a simple explanation for the findings. To address this, immunoprecipitations of extracts from cells expressing mutant p53 were examined. Immunoprecipitation with anti-p53 antibody (CM5) and subsequent immunoblotting with anti-Nrf1, did not detect a band with the migration size corresponding

to Nrf1 (Fig. EV3B). Immunoprecipitation with anti-Nrf1 antibody and subsequent immunoblotting with anti-53 (CM5) was less conclusive. Although a band with the molecular weight of p53 is detected with CM5 in these Nrf1 immunoprecipitates, it is unclear whether the intensity is above background. Further, the band intensity does not increase with the use of more extract. Taken

**Figure 5.  Mutant 172H p53 occupies genes and regulates their expression after X-radiation.**

(A) 24 genes are up and bound, and 30 genes are down and bound at 3 h after 9.5 Gy. Data is derived from N = 3 mice. Genes with an adjPval <0.1 in color. Statistical analysis for differential gene expression analysis was performed using DESeq2 (Love et al, 2014). (B) GSEA for the 24 up genes gave GO:BP terms. The number of shared genes that overlap with the indicated gene sets is shown in parenthesis. Statistical analysis for Gene Set Enrichment Analysis (GSEA) was performed using web tools provided by The University of California, San Diego and the Broad Institute (https://www.gsea-msigdb.org/gsea) (Mootha et al, 2003; Subramanian et al, 2005). (C) TRAP analysis gave top matrices. Statistical analysis for motif analysis was done by TRAP (Transcription Factor Affinity Prediction) using web tools provided by the Max Planck Institute for Molecular Genetics (http://trap.molgen.mpg.de) (Thomas-Chollier et al, 2011). (D) Using Nrf1 occupancy taken from an ENCODE data set, 22 out of the 24 genes had a detectable Nrf1 motif under their p53 ChIP peak. (E) Motifs that are detected in (D) match well with a known Nrf1 matrix. (F) Comparison with an Nrf1 ChIP-seq data set shows 22 of p53 peaks overlap with Nrf1 peaks. (G) 15 of the 24 genes are also occupied in ++ and all of the genes show occupancy with the heterozygous 172H/+. (H) RNA-seq data shows only one (Birc2) gene regulated in +/+ with 4 more genes regulated in m/+ bone marrow. Data is derived from N = 3 mice. Statistical analysis for differential gene expression analysis was performed using DESeq2 (Love et al, 2014). (I) ChIP-seq profile for Xpo5, Cbx5, and Msh3 at 3 h after X-radiation. (J) Corresponding RNA-seq data for the genes in (I) is shown. Statistical analysis for differential gene expression analysis was performed using DESeq2 (Love et al, 2014). Source data are available online for this figure.

together, there is little evidence to support the interaction between mutant p53 and Nrf1 at this time.

Basal gene expression was examined by RNA-seq using untreated bone marrows of the various genotypes (Appendix Fig. S5A). When comparing bone marrows from engineered mice to that of wild type, little differential gene expression was detected, except to some extent with 172H/−. Given the variation in LSK+ cell numbers across these genotypes (Fig. 1B), it might be expected that more substantial gene expression differences would be seen. Although only minimal differences were seen when comparing to wild type bone marrow, there are more substantial distinct gene expression when comparing the various 172H genotypes to that of the null (Appendix Fig. S5B,D). This is consistent with the idea that there is indeed a gain of gene expression upon expression of the mutant 172H p53 protein. GSEA on the differentially expressed genes in 172H/− bone marrow compared to null reveals enrichment for specific gene set terms for both up and down-regulated genes (Appendix Fig. S5C). It should be noted that p53 is a low scoring gene in the differential gene analysis comparing mutant to null bone marrow (Appendix Fig. S5B). This likely reflects that only a small subset of cell types in the bone marrow actually express p53 mRNA.

### The heterozygous 172H/+ genotype has an expanded transcriptome compared to wild type p53 +/+

The integrated RNA-seq and ChIP-seq analysis was then performed on the bone marrow of the 172H/+ mice after radiation treatment (Appendix Fig. S6A). In all, 87 genes were occupied and upregulated with 115 genes being occupied and downregulated. Each of these gene sets was examined for the p53-dependence of their expression by comparing to p53 null bone marrows (Appendix Fig. S6B–E). 7 upregulated and 13 downregulated genes were eliminated from the lists as those showed significant alteration after radiation in the p53 null samples (Appendix Fig. S6D). Thus, for genes that are strictly p53-dependent, eighty were occupied and upregulated in the 172H/+ bone marrow with a corresponding 102 genes being occupied and downregulated (Fig. 6A). GSEA shows Biological Process terms that are commonly seen with the wild type p53 response to radiation (compare Figs. 2B and 6B). When expression of these 80 up genes was examined in X-radiated bone marrows of the +/+ and 172H/− genotypes, it was found that the majority of these genes were activated in the +/+ setting (38 genes). Sixteen of these genes were also upregulated in the 172H/− bone marrow, with the remaining 19 being peculiar to the 172H/+

genotype (Fig. 6C). The genomic sequences underneath the peaks detected by ChIP-seq for these subsets of genes were subjected to TRAP analysis. For the genes shared with +/+ and those that are exclusive to 172H/+, the top detected matrices were that of wild type p53 (Fig. 6D). In contrast, the genes shared with 172H/− are enriched for other matrices, most notably that of Nrf1 (Fig. 6E). Two of the genes that are shared between the +/+ and172H/+ genotypes (i.e., Cdkn1a/p21 and Bbc3/Puma) show corresponding peaks with no detectable overlap with an Nrf1 ChIP-seq track (Fig. 6F). Both are upregulated in +/+ and 172H/+ (Fig. 6F). In contrast, a representative gene that is shared between 172H/+ and 172H/−, Zbtb41, shows a p53 ChIP peak for all genotypes as well as one from the Nrf1 ChIP-seq (Fig. 6F). Zbtb41 is upregulated in both mutant genotypes (172H/+ and 172H/−), but not the +/+ (Fig. 6F). Thus, the cistromic genes that are associated with the 172H/+ genotype appear to have two distinct modes of gene occupancy: one is through a canonical p53 motif and the other is through a non-canonical binding mechanism, possibly involving Nrf1.

RT-qPCR analysis was used to validate effects seen with RNA-seq. In the bone marrow, RT-qPCR for each genotype shows comparable effects as what is seen by RNA-seq for three targets (Cdkn1a, Bbc3, and Ccng1) (Fig. EV2A–F). RT-qPCR experiments show that there are targets that are only upregulated in the m/− bone marrow (Snora64, Fig. EV2G), but not in +/+ or m/+ tissue. Conversely, there is a target Mroh8 that is only upregulated in the m/+ bone marrow but not in either +/+ or m/− tissue (Fig. EV2H). This confirms the existence of a mutant p53-specific target (Snora64) as well as a target that is only seen in the heterozygous m/+ bone marrow, Mroh8. Both these targets show an overlap between the p53 ChIP peak and that of Nrf1 (Fig. EV2I).

### The presence of Mdm2 enhances wild type p53 gene occupancy on a subset of targets

RNA-seq analysis was performed for the radiation response in transgenic mice that have been engineered to overexpress Mdm2 (Fig. 7A) as is shown for the other examined genotypes (Fig. 1C). GSEA gives distinct gene set terms for this genotype (Fig. 7B) as compared to the others (Fig. 1D). Integration with the ChIP-seq data provides a small but significant set of genes that are differentially expressed in the Mdm2 transgenic mice after 9.5 Gy radiation treatment with 29 being occupied and upregulated and 19 being occupied and downregulated (Fig. 7C). For the 29 up genes, all show p53-dependence with substantial numbers also being

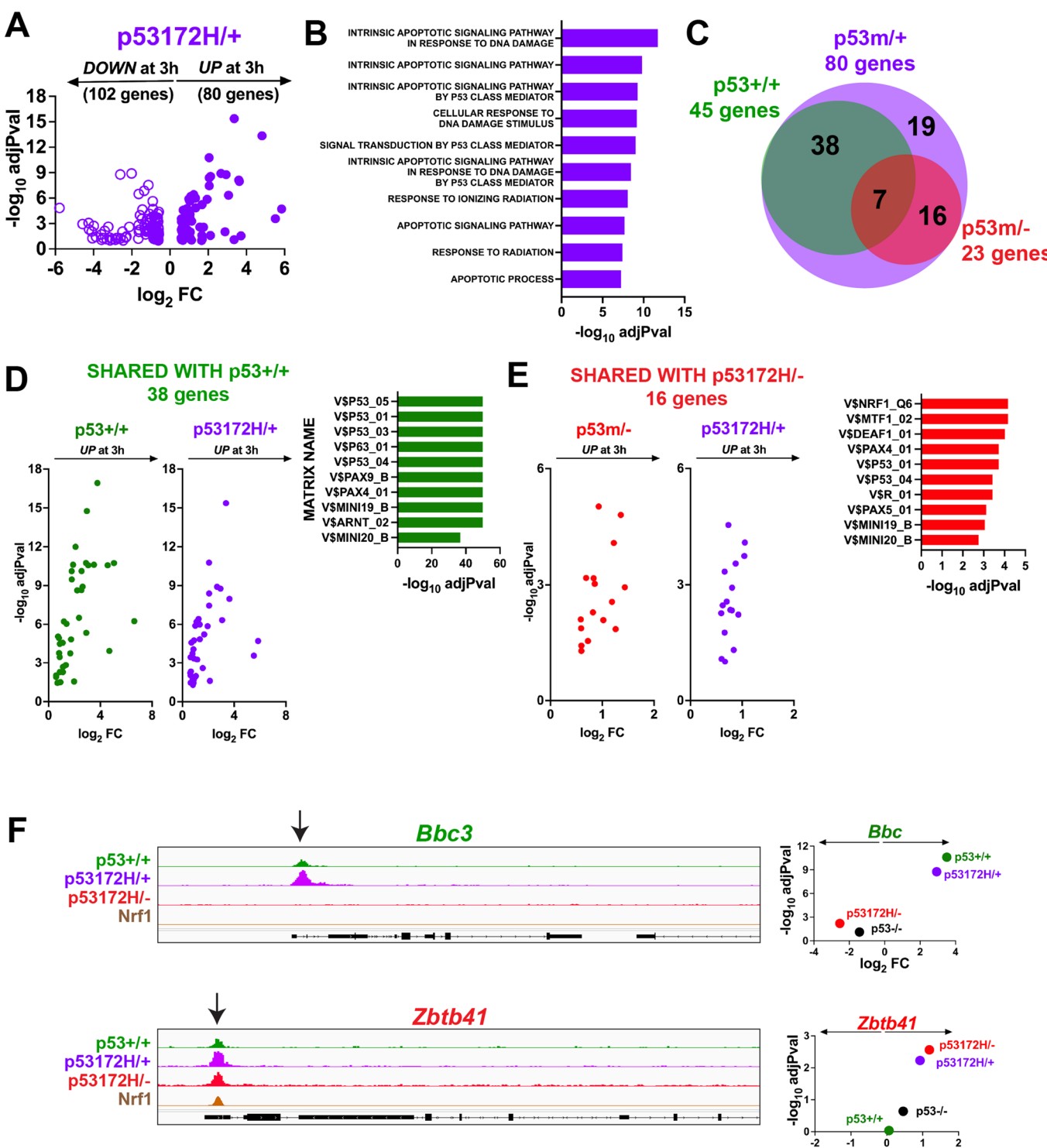

regulated in +/+ (19 genes) and 172H/+ (18 genes) bone marrows, less so with the 172H/− (9 genes) (Fig. 7D,E). It is interesting that there are 10 genes that are upregulated and occupied in the Mdm2 mice but are not seen as differentially expressed after radiation for the wild type bone marrows (Fig. 7E). ChIP-seq analysis for these 10 Mdm2-specific genes shows substantially overlap with mutant p53 occupancy as well as Nrf1 with a smaller subset showing

binding by wild type p53 (Fig. 7F). Comparison of the ChIP-seq peaks without regard for gene expression changes shows that there is a substantial number of sites in which expression of Mdm2 prevents occupancy (132 peaks), but remarkably a large number of sites (195 peaks) that are only detected in the presence of Mdm2 (Fig. 7G). Of these genes that are only regulated in the presence of Mdm2, *Birc2* is an example of one that is occupied both in the

**Figure 6.   The heterozygous 172H/+ genotype has an expanded transcriptome compared to wild type p53 + /+.**

(A) 80 genes are up and bound, and 102 genes are down and bound at 3 h after 9.5 Gy. Data is derived from $N = 3$ mice. Genes with an adjPval <0.1 are in color. These all do not show regulation in p53 null bone marrow and are hence considered p53-dependent in their expression (See Appendix Fig. 6). Statistical analysis for differential gene expression analysis was performed using DESeq2 (Love et al, 2014). (B) GSEA on the 80 up genes gave indicated GO:BP terms. Genes that overlap with indicated gene sets are shown. Statistical analysis for Gene Set Enrichment Analysis (GSEA) was performed using web tools provided by The University of California, San Diego and the Broad Institute (https://www.gsea-msigdb.org/gsea) (Mootha et al, 2003; Subramanian et al, 2005). (C) Of the 80 genes identified in (A), 45 are upregulated in p53 + /+ and 16 in p53m/−. Genes that overlap are shown as a Venn diagram. (D, E) Two subsets in these 80 upregulated genes: 38 genes are up in p53 + /+, and 16 are also up in the 172H/− mice. TRAP analysis detected top matrices. Data is derived from $N = 3$ mice. Statistical analysis for differential gene expression analysis was performed using DESeq2 (Love et al, 2014). Statistical analysis for motif analysis was done by TRAP (Transcription Factor Affinity Prediction) using web tools provided by the Max Planck Institute for Molecular Genetics (http://trap.molgen.mpg.de) (Thomas-Chollier et al, 2011). (F) On the left, ChIP-seq profiles at 3 h after Xray. Nrf1 occupancy is taken from an ENCODE data set for MEL cells (Accession: ENCSR135SWH). On the right, RNA-seq data for these genes. Statistical analysis for differential gene expression analysis was performed using DESeq2 (Love et al, 2014). Source data are available online for this figure.

absence and presence of Mdm2 while Sp1 only shows occupancy in the presence of Mdm2 (Fig. 7H). These results provide an unexpected finding. Contrary to current thinking, it appears that expression of Mdm2 can enhance wild type p53-dependent gene occupancy and expression after radiation treatment under these conditions in the bone marrow.

## Discussion

Previous studies suggested that the dominant-negative effect of mutant p53 as well as the negative regulation by Mdm2 occurs regardless of the specific target. In contrast, here is shown that both mutant p53 and Mdm2 exert gene-selective effects. As a confirmation of the biological relevance of the experimental model, wild type p53 is shown to occupy and upregulate a defined set of genes that have been previously well characterized as bona fide p53 targets (Figs. 2A and 3C). These are clearly p53-dependent as the upregulation is not seen in bone marrows that either do not express p53 (−/−) or only express the 172H mutant protein (172H/−) (Fig. 3A). Previous studies have indicated that the C-terminus plays a key negative regulatory role in the bone marrow responses to radiation by wild type p53 (Wang et al, 2011). Here, it is shown that this effect can be localized to the terminal 19 amino acids (Fig. 4). Based on published studies, it was expected that co-expression of either mutant p53 (172H/+) or the negative regulator Mdm2 (Mdm2/Mdm2) would suppress this wild type p53-regulated gene expression. Surprisingly, expression of only six of the 20 genes was affected (Fig. 3B,C). The majority of the genes, many of which are well known p53 targets, maintain their gene expression after radiation even in the presence of the two putative negative regulators. Regardless of effects on gene expression, the occupancy of these genes by wild type p53 is unaffected (Fig. 3E,F), arguing for a mechanism that occurs after DNA binding. Nevertheless, these results argue that under these experimental conditions, mutant p53 does not exert a dominant-negative effect on many wild type p53 targets. Likewise, Mdm2 does not negatively regulate the ability of wild type p53 to do so either. The molecular basis for these gene selective effects of mutant p53 and Mdm2 remains to be elucidated. It is interesting that the effects on gene regulation are similar (Fig. 3C) between mutant p53 and Mdm2 suggesting a possible shared mechanism of action. The findings also beg the question of whether Mdm2 influences mutant p53 and its transcriptional program, a topic for future further study.

In vitro and cell-based analyses have shown that tumor-associated hotspot mutants of p53 lack intrinsic DNA binding and fail to occupy wild type p53 binding sites that conform to the well characterized consensus (Bargonetti et al, 1991; Bargonetti et al, 1993; Cho et al, 1994; el-Deiry et al, 1992; Joerger et al, 2006; Joerger et al, 2005; Kato et al, 2003; Kennedy and Lowe, 2022; Kern et al, 1991a; Kern et al, 1991b; Pfister and Prives, 2017). In contrast, the bone marrow model used here reveals an overlap in gene occupancy between wild type p53 and the 172H mutant (Figs. 3D and 5G). As expected, the 172H mutant fails to occupy and thus regulate a set of well characterized wild type targets (Fig. 3A,D). However, there are five wild type targets that are also occupied by the mutant, although there is no regulation of their gene expression (Fig. 3D). More remarkably, mutant 172H p53 occupies and upregulates 24 genes after radiation (Fig. 5A). Of these, the majority (14 genes) are also occupied by wild type p53, albeit not regulated (Fig. 5G). None of the ChIP peaks that are shared in their occupancy by wild type and mutant p53 appear to contain genomic sequences that match the well characterized wild type p53 consensus (Fig. 5C). Thus, there appears to be a non-canonical mode of DNA binding that is shared between wild type and the 127H mutant.

Detection of motifs under the ChIP peaks which show occupancy by wild type p53 but not the 172H mutant readily identifies known p53 consensus sequences (Fig. 2C,E). In contrast, similar analysis for ChIP peaks associated with mutant 172H-regulated genes does not present p53 matrices, but rather the top hit was for an Nrf1 matrix (Fig. 5C). Use of existing ENCODE data for mouse ChIP-seq of Nrf1 confirms that many of these 172H-regulated genes show a completer overlap between the mutant p53 ChIP peak and that of Nrf1 (Fig. 5F,G). 14 of these mutant p53-regulated genes also show occupancy with wild type p53 and Nrf1 motifs being identified under all of these ChIP peaks (Fig. 5D,E). The five wild type p53-regulated genes that also show mutant p53 occupancy similarly show Nrf1 occupancy and have identifiable Nrf1 motifs under their peaks (Fig. 3D). This presents the intriguing hypothesis that both wild type and mutant p53 can both occupy genes which have Nrf1 motifs, and possibly the basis for this DNA binding is via the Nrf1 transcription factor. It should be noted that there are a remaining 8 genes that are occupied and upregulated by mutant p53 that are not associated with wild type p53 (Fig. 5G).

Use of the 172H/+ genotype to further explore the interplay between mutant and wild type p53 reveals that such bone marrows retain much of the wild type p53 gene occupancy and regulation (Fig. 6A,D). However, the p53-dependent transcriptome in these tissues has expanded to now include genes that are also regulated by mutant p53 in the absence of wild type (172H/− genotype) (Fig. 6A,E). This suggests that the oncogenic activity that is seen in the m/+ genotype relies less on a dominant-negative effect and more on the expansion of gene regulation to include mutant p53

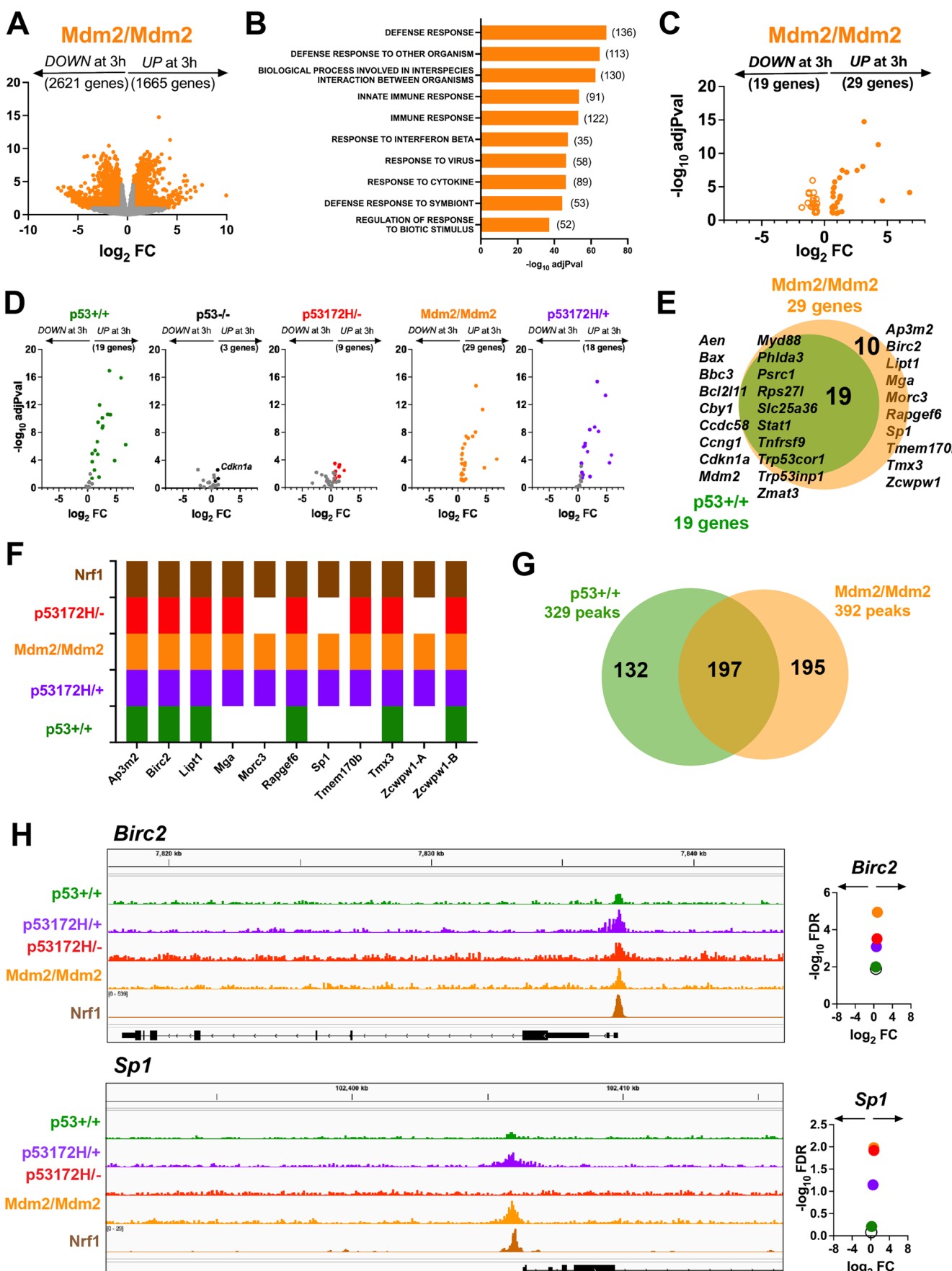

**Figure 7.  The presence of Mdm2 enhances wild-type p53 gene occupancy on a subset of targets.**

(A) 8 week-old mice of the indicated genotype were untreated or treated with 9.5 Gy of X-ray. The Mdm2 transgenic mice (Mdm2/Mdm2) contain a BAC construct containing the mouse *Mdm2* gene that is integrated in a single site as multiple copies (Jones et al, 1998). After 3 h, RNA was extracted from bone marrow, and subjected to RNA-seq analyses. Volcano plots are shown as -$\log_{10}$ adjusted *P* value (adjPval) versus $\log_2$ Fold change (FC). Data is derived from *N* = 3 mice. Differentially expressed genes upon radiation with an adjusted *P* value (False Discovery Rate) <0.1 and an absolute fold change >1.5 are shown as colored dots. Statistical analysis for differential gene expression analysis was performed using DESeq2 (Love et al, 2014). (B) GSEA analysis was performed for the significant differentially expressed up-regulated genes and the top Gene Ontology Biological Processes terms are shown with corresponding $-\log_{10}$ adjPval. Statistical analysis for Gene Set Enrichment Analysis (GSEA) was performed using web tools provided by The University of California, San Diego and the Broad Institute (https://www.gsea-msigdb.org/gsea) (Mootha et al, 2003; Subramanian et al, 2005). (C) 29 genes are up and bound, and 19 genes are down and bound at 3 h after 9.5 Gy. Data is derived from *N* = 3 mice. Genes with an adjPval <0.1 in color. Statistical analysis for differential gene expression analysis was performed using DESeq2 (Love et al, 2014). (D) The differential gene expression of these 29 up and bound genes in the indicated genotypes is shown. Data is derived from *N* = 3 mice. Statistical analysis for differential gene expression analysis was performed using DESeq2 (Love et al, 2014). (E) Of the 29 genes in (C), 19 are also upregulated in the p53 + /+ mice with 10 genes only regulated in the Mdm2/Mdm2 mice. (F) For each of the 10 upregulated genes in only Mdm2/Mdm2 bone marrow, the presence of a corresponding ChIP peak for each indicated genotype is shown. Nrf1 occupancy is taken from an ENCODE data set for MEL cells (Accession: ENCSR135SWH). (G) Of the 329 peaks which show p53 occupancy in p53 + /+ mice (Fig. 1E), 197 overlap with the 392 peaks found in the Mdm2/Mdm2 mice, with 195 peaks only being found in the Mdm2/Mdm2 mice. (H) On the left, ChIP-seq profiles at 3 h after Xray. Nrf1 occupancy is taken from an ENCODE data set for MEL cells (Accession: ENCSR135SWH). On the right, RNA-seq data for these genes. Source data are available online for this figure.

occupied and regulated genes. Indeed, the early tumorigenesis studies using the 172H/+ mice showed that these had an altered cancer spectrum and increased metastases that was distinct from the p53 null mice (Lang et al, 2004; Olive et al, 2004). If the driving mechanism here were a dominant-negative effect of the mutant on wild type, it would not have been expected to see such differences between the heterozygous (172H/+) and null (−/−) genotypes.

If one compares the genes that are occupied and regulated in the 172H/− bone marrow (Fig. 5A) to that of 172H/+ bone marrow (Fig. 6A), there are only two genes that overlap: *Nr1d2* and *Zcwpw1*. The data showing this shared regulation and occupancy for *Zbtb41* is shown in Fig. 6F. Further, of the 80 such occupied and upregulated genes in 172H/+, only 16 of these are also upregulated in the 172H/− bone marrow (the latter not necessarily being occupied by p53) (Fig. 6C,E). It is unclear occupancy in the 172H/ − mice for these 16 shared genes is not detected. The studies in bone marrow have been challenging and this may reflect the limitations of the ChIP-seq experiments. The data is consistent with the idea that there are transcriptional programs that are shared between 172H/+ and 172H/− as well as gene sets that are unique to each genotype.

Elucidating underlying molecular mechanisms for the ability of mutant p53 to execute a transcriptional program and how this compares to that of wild-type is clearly the long-term goal. It was necessary to establish first that mutant p53 does exert transcriptional effects that are associated with occupancy of genomic sites. The next important steps will include ChIP-seq analyses using antibodies to appropriate post-translational modifications of histones and determination of mutant p53-dependent effects on chromatin state. There is a growing body of evidence that not all tumor-derived mutant p53 proteins have similar activities. The translational potential of findings with mutant p53 thus will rely on teasing out general versus mutant-specific effects. It remains to be seen whether these findings can be generalizable or are specific to 172H. Further, determining the relevance of these studies in humans and ascertaining whether findings in mice have translational potential in human patients remains to be seen.

Previous studies have clearly demonstrated a critical role for p53 in the short- and long-term responses to radiation in vivo. The acute response to DNA damage in vivo, upon exposure to ionizing radiation, is strictly p53-dependent (Gudkov and Komarova, 2003; Komarova et al, 1997; Komarova et al, 2000; Komarova et al, 2004;

Lowe et al, 1993; MacCallum et al, 1996; Song and Lambert, 1999). Long-term sensitivity to lethal doses of radiation is likewise dependent on p53 with both p53 null- and mutant p53-expressing mice surviving at doses which kill wild type mice (Gudkov and Komarova, 2003; Komarova et al, 2004). In competitive repopulation experiments in mice, non-irradiated hematopoietic stem cells (HSCs) can outcompete irradiated wild type HSCs in a p53-dependent manner (Bondar and Medzhitov, 2010). HSC numbers are regulated by p53, being increased in both p53 null and mutant p53 bone marrows (Akala et al, 2008; Asai et al, 2011; Chen et al, 2008; Liu et al, 2009; Pant et al, 2012; TeKippe et al, 2003). There is also evidence that mutation of p53 enhances HSC fitness over mere deletion as R248W p53 conferred a competitive advantage following transplantation of murine HSCs and promoted their expansion after both radiation (Chen et al, 2019) and chemotherapy-induced stress (Chen et al, 2018). This crucial role for p53 in radiosensitivity and regulation of HSC levels has been confirmed using mice expressing either the 172H p53 (the murine equivalent of the hotspot 175H) or overexpressing the negative regulator Mdm2 (Fig. 1A,B). Further, the key role that the bone marrow plays in the survival of these mice is demonstrated by the ability to reconstitute lethally irradiated mice with wild type tissues (Fig. 4B).

Despite recent advances in the treatment of cancer patients, the existence of a *TP53* mutation is still associated with a dismal outcome, in particular in hematologic malignancies (Bally et al, 2014; Bejar et al, 2014). In vitro and in vivo studies however have provided alternative models for the effect of missense mutation of one p53 allele when expressed in the heterozygous state with a remaining wild type allele. In some cases, mainly through cell-based studies, it appears that mutant p53 can exert a dominant-negative effect on the wild-type protein activities (Boettcher et al, 2019; Giacomelli et al, 2018). In other experimental models including the use of genetically engineered mice, the mutant p53 has oncogenic activities not seen with mere p53 loss (Bougeard et al, 2008; McBride et al, 2014; Zerdoumi et al, 2013). Although several modes of action have been explored, the underlying molecular basis for these enhanced tumorigenic phenotypes with mutant p53 is likely to be highly cancer type dependent. Accordingly, the observations herein are somewhat unconventional. Not only is there apparent transcriptional activity for mutant p53 in response to DNA damage, it appears that there may be a cooperative effect by the

mutant on wild type p53 functions. These novel findings need broad validation with other missense mutants. Moreover, future studies should address responses to lower doses of radiation that mirror chronic DNA damage found in tumorigenesis. In addition, given the cellular heterogeneity of the bone marrow, a single-cell RNA-sequencing approach would offer a more refined description of cell-specific responses to radiation in *Trp53*-mutated bone marrows.

Taken together, these findings indicate that wild type p53 may have non-canonical means for regulating gene expression that it shares with a tumor-derived mutant p53. The notion of "gain-of-function" oncogenic activity has been conceptually challenging. The results here indicate that mutant and wild type p53 may share biochemical activities, and that the more accessible idea may be that there is selective loss of function upon missense mutation of p53. This is a notion proposed by others (Kennedy and Lowe, 2022) which is now given some experimental support. This has important implications in the prognosis of mutant p53 expression human cancers as well as in the development of targeted therapies. Given the apparent importance of p53 in human cancer, it has been disappointing that there has been limited progress to date in this area. A more complete understanding of the overlap in wild and mutant p53 activities will be necessary to address this crucial issue.

# Methods

### Reagents and tools table

| Reagent/resource | Reference or source | Identifier or catalog number |
|---|---|---|
| **Experimental models** | | |
| C57BL.6J p53 + /+ mice | Jackson Laboratories | JAX# 000664 |
| p53−/− mice | Jackson Laboratories | JAX# 002101 |
| p53172H/+ mice | Jackson Laboratories | JAX# 034619 |
| Mdm2/Mdm2 mice | Jones et al, 1998 | |
| p53Δ10/Δ10 mice | This study | |
| p53Δ19/Δ19 mice | This study | |
| 378 mouse esophageal squamous carcinoma cell line | Tang et al, 2021 | |
| **Antibodies** | | |
| Rabbit anti-mouse p53 | Leica Microsystems | p53-CM5P-L, Lot 6079173 |
| Normal mouse IgG | Santa Cruz Biotechnology | sc-2025, Lot F1819 |
| Mouse antibody to p21 | BD Pharmingen | B556431 |
| Rabbit monoclonal antibody to PUMA | Cell Signaling | 14570S |
| Mouse monoclonal antibody to Beta-actin | Sigma | A5316 |
| Alexa 488 Donkey anti-rabbit IgG | Jackson Immuno Research | 711-545-152 |

| Reagent/resource | Reference or source | Identifier or catalog number |
|---|---|---|
| Goat anti-mouse AlexaFluor 680 | Invitrogen | A21057 |
| Rabbit polyclonal to Nrf1 | Abcam | Ab34682 |
| Rat anti-CD16/32 | eBioscience | Clone 93, #14-0161-82 |
| PE-conjugated anti-mouse CD117/cKit | eBioscience | Clone 2B8, #12-1171-82 |
| FITC-conjugated anti-mouse lineage cocktail (B220, CD3, CD11b, Gr1, Ter119) | eBioscience | #50-112-3250 |
| APC-conjugated anti-mouse Sca1/Ly6a | eBioscience | Clone D7, #17-5981-82 |
| **Oligonucleotides and other sequence-based reagents** | | |
| *Cdkn1a* Forward RT-qPCR Primer | 5'-GCAGACCAGCCTGACAG ATT-3' | |
| *Cdkn1a* Reverse RT-qPCR Primer | 5'-CCTGACCCACAGCAGAA GAG-3' | |
| *Bbc3* Forward RT-qPCR Primer | 5'-CGGCGGAGACAAGAA GAG-3' | |
| *Bbc3* Reverse RT-qPCR Primer | 5'-CTCCAGGATCCCTGGGT AAG-3' | |
| *Ccng1* Forward RT-qPCR Primer | 5'-GGCTTTGACACGGAGAC ATT-3' | |
| *Ccng1* Reverse RT-qPCR Primer | 5'-GGATCAAATCAGTCGCC AGT-3' | |
| *Gapdh* Forward RT-qPCR Primer | 5'-AAGGGCTCATGACCACA GTC-3' | |
| *Gapdh* Reverse RT-qPCR Primer | 5'-GGATGCAGGGATGATGT TCT-3' | |
| *Mroh8* Forward RT-qPCR Primer | 5'-CCTGTATCCAGCAGACA GCA-3' | |
| *Mroh8* Reverse RT-qPCR Primer | 5'-ATGGCATAAAAGGTGGC TGA-3' | |
| *Snora64* Forward RT-qPCR Primer | 5'-TCTGAAAATCATGCCAG TGC-3' | |
| *Snora64* Reverse RT-qPCR Primer | 5'-GTGGTGCTCCTCTAGGC AAG-3' | |
| **Chemicals, enzymes and other reagents** | | |
| Protease Inhibitor Cocktail Set III | EMD Millipore | |
| Protein A-Sepharose | Amersham BioSciences | |
| 0.5% BSA | Fisher Scientific | #9048-46-8 |
| RNAeasy Mini Kit | Qiagen | 74104 |
| NEBNext Ultra RNA Library Prep Kit for Illumina | New England BioLabs | #E7530S/L |
| RNAlater | Thermo Fisher Scientific | |
| RNAeasy Mini Kit | Qiagen | 74104 |
| Quanta qScript cDNA Supermix | VWR | |
| Power-SYBR Green PCR Master Mix | Thermo Fisher-Applied Biosystems | 4367659 |

 

| Reagent/resource | Reference or source | Identifier or catalog number |
|---|---|---|
| 1% formaldehyde/PBS | Millipore Sigma | FX0415 |
| EDTA | Thermo Fisher Scientific | #78442 |
| Protein A Dynabeads | Invitrogen | #10002D |
| QIAquick PCR purification kit | Qiagen | 28104 |
| Qubit dsDNA HS Assay Kit | Life Technologies | #Q32851 |
| NEBNext ChIP-Seq Library Prep Master Mix Set for Illumina | New England BioLabs | #E6240S/L |
| **Software** | | |
| GraphPad Prism | GraphPad Software | v.10.4.1 for MacOS |
| GSEA | Mootha et al, 2003; Subramanian et al, 2005 https://www.gsea-msigdb.org | |
| TRAP | Thomas-Chollier et al, 2011 http://trap.molgen.mpg.de | |
| Integrative Genomics Viewer | Robinson et al, 2011 https://software.broadinstitute.org/software/igv | v.2.6.3 |
| WebLogo 3 | Crooks et al, 2004 https://weblogo.berkeley.edu | |
| R | https://www.r-project.org/ | v3.6-4.0 |
| Bioconductor | http://www.bioconductor.org | 3.2 |
| csaw | Lun & Smyth, 2016 https://bioconductor.org/packages/release/bioc/html/csaw.html | |
| Rsubread | Liao et al, 2019 https://bioconductor.org/packages/release/bioc/html/Rsubread.html | |
| ChIPpeakAnno | Zhu et al, 2010 | |
| EnsDb.Mmusculus | https://bioconductor.org/packages/EnsDb.Mmusculus.v79/ | v.79 |
| **Other** | | |
| Irradiator | Precision X-ray | X-RAD 32 |
| Cryo-Cup Grinder with Pestle | Research Products Int'l Corp | |
| Nitrocellulose Membranes | BioRad | |
| Odyssey CLx | LI-COR | |
| BD LSRFortessa™ Cell Analyzer | BD Biosciences | |
| Illumina NextSeq 500 High-throughput Benchtop Sequencer | Illumina | |
| Applied Biosystems Step One Plus Real-Time PCR System | Applied Biosystems | |
| Bioruptor Plus Sonication Device | Diagnode | #B01020001 |
| DynaMag-2 Magnet | Invitrogen | #123.21D |
| Qubit 2.0 Fluorometer | Life Technologies | #Q32866 |

## Mice

All experiments in mice were performed in accordance with a protocol approved by the Institutional Animal Care and Use Committee (IACUC-2015-0082, protocol no. 14-1378 CONT). All studies were performed on male mice. The mutant p53 mice have been backcrossed into a C57BL.6J background. The Mdm2 transgenic mice (Mdm2/Mdm2) were a kind gift of Stephen Jones (Frederick National Laboratory) (Jones et al, 1998). These mice contain a BAC construct containing the mouse *Mdm2* gene that is integrated in a single site as multiple copies. Two novel p53 transgenic mouse models were generated where the mouse p53 C-terminal 10 (ATG) and 19 (CAG) amino acids were converted to STOP (TGA) codons by CRISPR/Cas9 editing in single cell mouse embryos. Cas9 protein was honed to exon 11 region of p53 using optimized guide RNAs (gRNA). A double-stranded break was induced by Cas9 leading to subsequent homology-directed repair with a single-stranded donor template. The template used for Δ10 contains an ATG → TGA at M381 and the template used for Δ19 contains a CAG → TGA at Q372. Targeting gRNA and single-stranded repair oligonucleotides were synthesized, designed, and validated by the Genome Engineering & iPSC Center at Washington University. Pronuclear injections to derive the Trp53Δ10 (p53Δ10/Δ10) and Trp53Δ19 (p53Δ19/Δ19) were performed by the Mouse Genetics and Gene Targeting Center at the Icahn School of Medicine at Mount Sinai. Sanger sequencing of tail DNA confirmed gene editing. These mice were derived in a C57BL/6J background. The Δ10 and Δ19 mice are available upon request.

## Irradiation treatment

Adult 6- to 8-week-old C57BL/6 mice of varying genotypes were exposed to lethal radiation in a Precision X-Ray (X-RAD 32). Pie chambers were used for immobilization and the total body radiation was performed with no shielding. A total X-ray dose of 9.5 Gy was administered for RT-qPCR, RNA-seq, and ChIP-seq experiments, and tissues were collected three hours after irradiation. Mice were sacrificed by regulated exposure to $CO_2$ for 4 min, compliant with the AVMA $CO_2$ Euthanasia Guidelines. For the survival experiments, mice received a lethal dose delivered in two equal fractions separated by 4 h.

## Immunoblotting

Upon dissection, spleen and thymus were snap frozen in Liquid Nitrogen and stored in foil at $-80\,°C$ until extracts were made. Thymus and spleen were pulverized in Liquid Nitrogen using a Cryo-Cup Grinder with Pestles (Research Products Int'l Corp.) Either the powdered organ sample or a single cell suspension of bone marrow were added to TPER protein extraction buffer containing a 1:100 dilution of Protease Inhibitor Cocktail Set III (EMD Millipore) and lysed on ice for 10–15 min before spinning out the cellular debris. Typically, 50 μg protein are separated on a 10% or 12% SDS polyacrylamide gel before transfer onto nitrocellulose membranes (BioRad). The blocked membranes (1x Phosphate Buffered Saline containing 0.1% Tween 20 and 1% Non-

Fat Dry Milk) are immunoblotted with the appropriate primary antibody and incubated at 4 °C overnight. After washing at room temperature with blocking buffer, the membranes are incubated with the appropriate AlexaFluor conjugated secondary antibody in a covered container for 1 h. The membranes are washed in 1× PBST and visualized on a LI-COR Odyssey CLx.

## Immunoprecipitations

378 cells are derived from the metastasis of a mouse esophageal squamous carcinoma with the genotype p53175H/− (Tang et al, 2021). Cells were washed and harvested in PBS, then lysed as described for immunoblotting. In total, 500 μg of proteins were mixed with primary antibody and 30 μl of 50% slurry of protein A-Sepharose (Amersham BioSciences) and rocked at 4 °C for 2 h. The beads were washed 3 times with RIPA (5 mM Tris-HCl pH 8.0, 150 mM NaCl, 1% NP-40, 0.5% deoxycholate, 0.1% SDS, 5 mM EDTA), resuspended in 30 μl of 2× sample buffer, and heated to 95 °C for 5 min. Proteins were resolved on a 10% or 12% polyacrylamide gel and analyzed by immunoblotting.

## Isolation of mouse bone marrow cells for flow cytometry analysis

Cells were isolated by flushing bone marrows from mouse femora and tibiae. In all, $2 \times 10^6$ cells were spun down, and then resuspended in 100 μL of Dulbecco's PBS supplemented with 0.5% BSA and rat anti-CD16/32 IgG antibody (1:100). $0.5 \times 10^6$ cells were separately aliquoted in a 96-well flat-bottom plate and stained with the following fluorescent anti-mouse antibodies (all dilutions at 1:500): PE-conjugated anti-mouse CD117/cKit; FITC-conjugated anti-mouse lineage cocktail (B220, CD3, CD11b, Gr1, Ter119) and APC-conjugated anti-mouse Sca1/Ly6a. Flow cytometry analyses were carried out on the BD LSRFortessa™ Cell Analyzer (BD Biosciences).

## RNA-sequencing (RNA-seq)

RNA was extracted in biological triplicate samples from single-cell suspensions of bone marrow using the RNeasy Mini Kit. RNA-Seq libraries were prepared from 1 μg total RNA using the NEBNext Ultra RNA Library Prep Kit for Illumina following the manufacturer's instructions and sequenced on the Illumina NextSeq 500 high-throughput benchtop sequencer (sequencing chemistry: 75 bp single-end reads, >35 M reads per sample). All RNA-Seq samples went through extensive quality control during preprocessing. The reference annotation used was Grcm38 Gencode M25. Exploratory analysis was done using dimensionality reduction techniques. Principal Component Analysis (PCA) plots were generated using the top 500 genes with the highest variation in gene expression across samples. Differential expression analysis was performed using DESeq2 (Love et al, 2014). Log$_2$ fold changes (lfc) were shrinked by using apeglm estimator. Genes with P-adjusted value < 0.1 and absolute (lfc) ≥0.58 were considered as differentially expressed. Volcano plots were generated using Prism software. Gene Set Enrichment Analysis (GSEA) was performed using web tools provided by The University of California, San Diego and the Broad Institute (https://www.gsea-msigdb.org/gsea) (Mootha et al, 2003; Subramanian et al, 2005).

## Reverse transcription-quantitative PCR (RT-qPCR)

Organ pieces were immediately placed in RNAlater during dissection and stored at 4 °C until RNA is extracted using RNeasy Mini Kit and following manufacturer's instructions. The isolated RNA was quantitated and 1 μg RNA was converted to cDNA using Quanta qScript cDNA supermix (VWR). Quantitative PCR using a reaction mix containing 1:40 dilution of each cDNA, 10 μM paired primers and Power-SYBR Green 2× Master Mix (Applied Biosystems) was performed on an Applied Biosystems Step One Plus Real-Time PCR system. The program was run for 40 cycles at 60 °C. The relative gene expression when normalized to GAPDH is calculated using the ddCt algorithm. See Regents & Tools Table for primer sequences.

## Chromatin immunoprecipitation with sequencing (ChIP-seq)

Single-cell suspensions from mouse bone marrow were cross-linked in 1% formaldehyde/PBS solution. The reaction was quenched with 0.125 M Glycine. Cell pellets were lysed in RIPA buffer containing a 1:100 dilution of protease and phosphatase inhibitor cocktail with EDTA (Thermo Fisher Scientific). Lysates were sonicated at 4 °C (35 cycles, 30 s ON/30 s OFF) using Bioruptor Plus sonication device (Diagenode) to produce DNA fragments that are approximately 200–500 bp. The sonicated lysates were cleared by centrifugation (16,000 × g, 10 min, 4 °C). A consistent amount of protein (generally 1 mg for the irradiated BMs; 5–6 mg for the leukemic spleens) was used for immunoprecipation. Lysates were precleared with Protein A Dynabeads (Invitrogen,) for 2 h at 4 °C using gentle rotation. The beads were separated using DynaMag-2 Magnet (Invitrogen) and samples were incubated with anti-p53 antibody or non-reactive mouse IgG antibodies (200 μg/0.5 mL) overnight at 4 °C using gentle rotation. The following day, samples were incubated with Protein A Dynabeads for 2 h at 4 °C using gentle rotation, then beads were separated from supernatants. A succession of 5-min washes of increased stringency (RIPA 2×, LiCl Buffer 5×, RIPA 2× and TE Buffer 2×) was performed at RT to ensure specificity. Finally, elution from the beads was accomplished at 65 °C for 10 min (Elution buffer: 50 mM Tris pH 8.0, 1% SDS, 10 mM EDTA). The elution step was repeated, and the supernatants were combined. Similarly, the input samples were brought to the same volume as the IP samples with elution buffer. To reverse the crosslinks, 5 M NaCl was added to samples to a final concentration of 200 mM. The pooled eluates and Input DNA were then incubated at 65 °C overnight. DNA was recovered the following day by column chromatography using QIAquick PCR Purification Kit (QIAGEN) in Buffer EB (10 mM Tris–HCl pH 8.5). DNA concentration was measured with the Qubit 2.0 Fluorometer (Life Technologies) using the Qubit dsDNA HS Assay Kit (0.2–100 ng). The total amount of recovered DNA ranged from 4 to 10 ng for BM samples, and from 15 to 35 ng for leukemic spleens. ChIP-Seq libraries were prepared using the NEBNext ChIP-Seq Library Prep Master Mix Set for Illumina (New England BioLabs) following the manufacturer's instructions and sequenced on the Illumina NextSeq 500 high-throughput benchtop sequencer (sequencing chemistry: 40 bp paired-end reads, >15 M reads per sample). The DNA binding repertoire of wild-type and mutant p53 was analyzed with Rsubread/csaw (Liao et al, 2019; Lun and Smyth,

2016). A sliding window was used to identify capture read counts between the tissue-specific p53 signal and the relevant input, using a window width of 50 bp. To define significant p53 peaks, a feature count matrix was filtered, globally normalized with dispersions calculated, following which a generalized linear model function was applied to estimate significant differences. To minimize the penalties for false discovery, immediately adjacent significant windows were merged (to a maximum width of 5 kb) and the final peak's *P* value was calculated on the merged and weighted windows. A significant peak was considered an independent test with a FDR *q*-value ≤ 0.05. The ChIPpeakAnno (Zhu et al, 2010) package was used to calculate the number of significant overlaps (minimum of 1 bp shared) of significant peaks across the different genotypes, with *P* value calculated from the findOverlapsOfPeaks function using a hypergeometric test. Significant ChIP-Seq peaks were also annotated to genes within a maximum distance of 10 kb from the transcription start site using EnsDb.Mmusculus.v79. To identify over-represented motifs, TRAP (Transcription Factor Affinity Prediction) web tools provided by the Max Planck Institute for Molecular Genetics (http://trap.molgen.mpg.de) (Thomas-Chollier et al, 2011). Consensus motifs were generated using WebLogo 3 (Crooks et al, 2004). Gene-annotated peaks were filtered with RNA-Seq DEGs to identify genes that were both significantly bound by wild-type or mutant p53 and significantly regulated by radiation. Peaks were visualized using the Integrative Genomics Viewer (Robinson et al, 2011).

## Statistical analysis

All experiments were performed with at least three biological replicates, as indicated, and were subjected to an unpaired two-tailed Student's *t* test or other appropriate statistical test as noted. Error bars represent standard error (SE) or standard error of the mean (SEM) as indicated. Significance values and number of biological replicates were as indicated in each figure legend.

## Data availability

RNA-seq and ChIP-seq data sets have been submitted to the Gene Expression Ominubus (Accession: GSE281318 and GSE281319). Nrf1 ChIPseq data is from an existing ENCODE data set for MEL cells (Accession: ENCSR135SWH).

The source data of this paper are collected in the following database record: biostudies:S-SCDT-10_1038-S44319-025-00375-y.

## Peer review information

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

## Acknowledgements

All mice were raised and maintained under standard conditions at the Center for Comparative Medicine and Surgery (CCMS) at Icahn School of Medicine at Mount Sinai. The services of several Shared Resource Cores at the Tisch Cancer Institute are gratefully acknowledged. Flow cytometry experiments were performed at the Flow Cytometry Core, the preparation of libraries and subsequent sequencing for the RNA-seq and ChIP-seq analyses were performed at the Cancer Genomics Technologies Core. Pronuclear injections to derive the Trp53Δ10 and Trp53Δ19 mice were performed by the Mouse Genetics Core. This work also relied on the advice of the Bioinformatics for Next Generation Sequencing (BiNGS) Core. All of these are supported in part by an NCI P30 Cancer Center Support Grant to the Tisch Cancer Institute (P30 CA196521). 378 cells were a kind gift from Anil Rustgi and Gizem Efe (Columbia University). The advice of Songhee Back, Dan Hasson, and Saul Carcamo are gratefully acknowledged. This work was supported by a grant from the National Cancer Institute to JJM (R01 CA257548).

## Author contributions

**Ramy Rahmé**: Conceptualization; Data curation; Formal analysis; Investigation; Methodology; Writing—original draft; Writing—review and editing. **Lois Resnick-Silverman**: Conceptualization; Validation; Investigation; Methodology. **Vincent Anguiano**: Conceptualization; Resources; Methodology. **Moray J Campbell**: Data curation; Formal analysis; Methodology. **Pierre Fenaux**: Conceptualization. **James J Manfredi**: Conceptualization; Formal analysis; Supervision; Funding acquisition; Investigation; Methodology; Writing—original draft; Project administration; Writing—review and editing.

Source data underlying figure panels in this paper may have individual authorship assigned. Where available, figure panel/source data authorship is listed in the following database record: biostudies:S-SCDT-10_1038-S44319-025-00375-y.

## Disclosure and competing interests statement

The authors declare no competing interests.

# Expanded View Figures

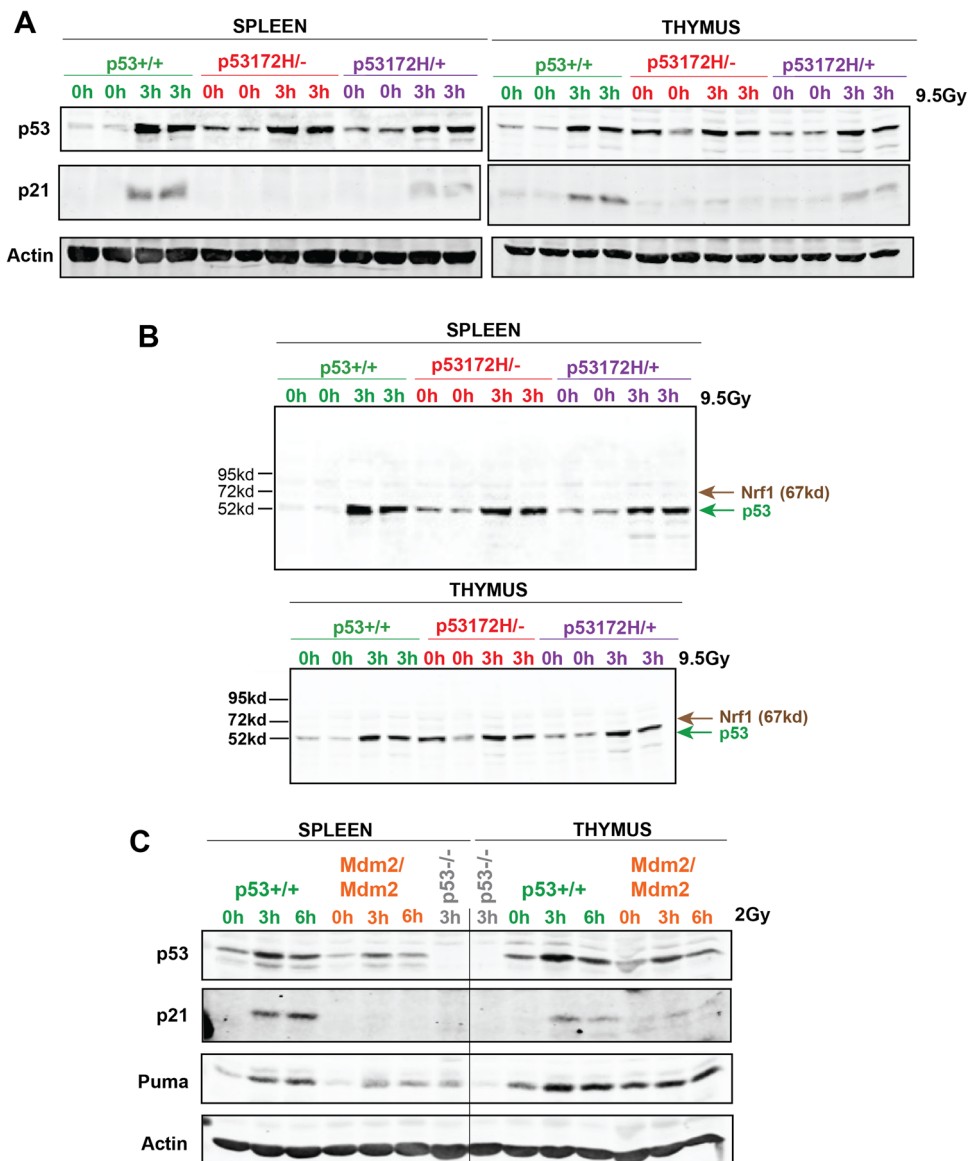

**Figure EV1.    p53 is detectable in spleen and thymus of mutant p53 and Mdm2-overexpressing mice.**

8-week old mice of the indicated genotype were untreated or treated with shown doses of X-ray. After the indicated time points, protein was extracted from spleen and thymus, and subjected to immunoblotting. Each lane represents tissue from a single animal. (**A**) Mutant p53 protein is induced by X-radiation in mouse spleen and thymus. Extracts from wild-type (p53 + /+), heterozygous mutant (p53172H/+) or hemizygous mutant (p53172H/−) spleen and thymus were immunoblotted with antibodies against either p53, p21, or Actin as indicated. (**B**) The antibody to p53 does not detect a band with the molecular weight of Nrf1 in mouse thymus or spleen. The immunoblots in (**A**) that were probed with the antibody to p53 prior to cropping are shown. The size of mouse Nrf1 (67kd) is indicated on the right. The position of standard molecular weight markers is shown on the left. (**C**) p53, p21, and Puma protein are induced by X-radiation in Mdm2 transgenic spleen and thymus. Extracts from wild-type (p53 + /+), transgenic Mdm2 (Mdm2/Mdm2), or p53 null (p53−/−) spleen and thymus were immunoblotted with antibodies against either p53, p21, Puma, or Actin as indicated.

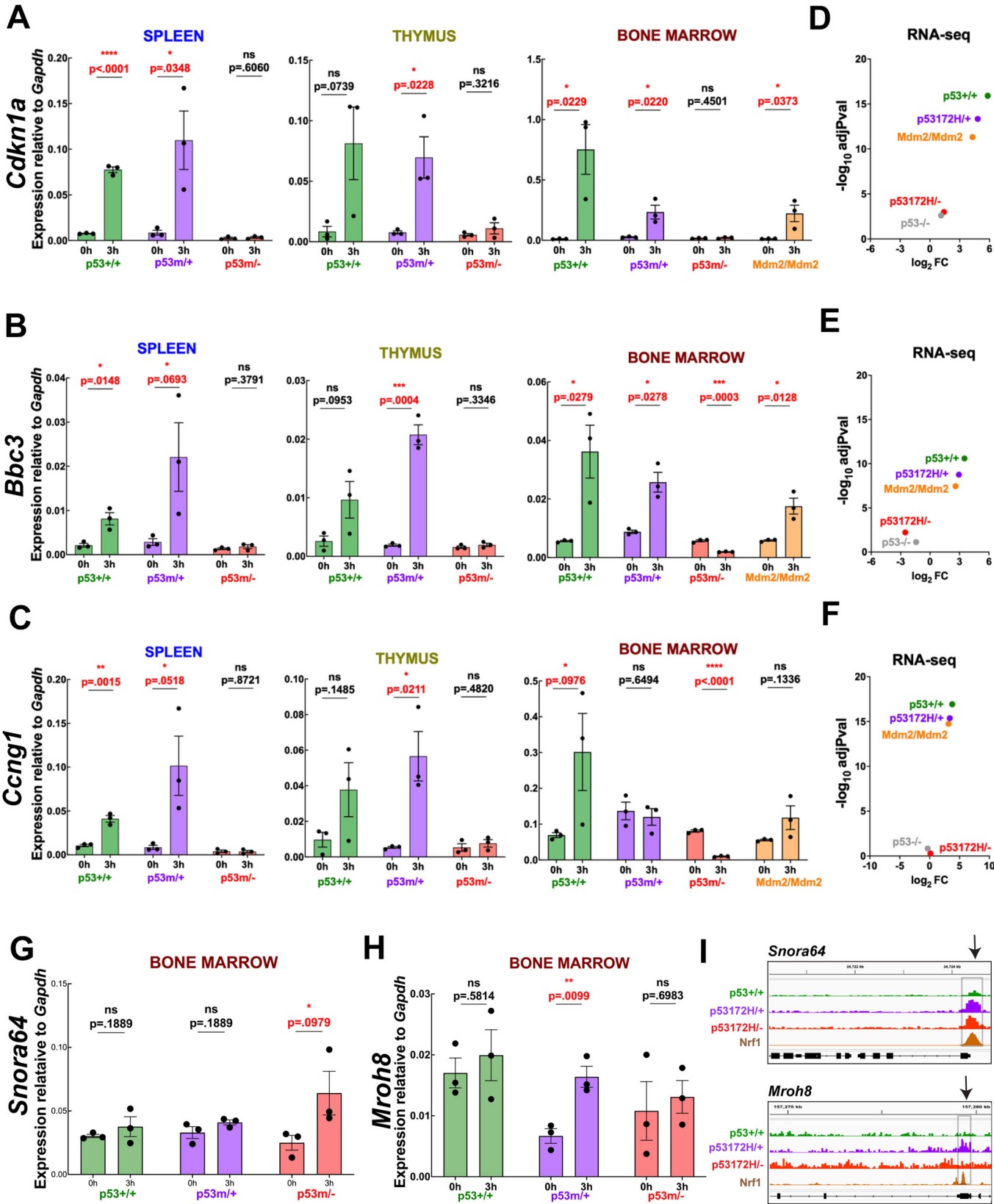

Figure EV2.  **RT-qPCR confirms gene expression changes observed by RNA-seq.**

(A–C) 8 week old mice of the indicated genotype were untreated or treated with 9.5 Gy of X-ray. After 3 h, RNA was extracted from spleen, thymus, and bone marrow, and subjected to RT-qPCR. Expression of *Cdkn1a* (**A**), *Bbc3* (**B**), and *Ccng1* (**C**) were determined relative to *Gapdh*. Each circle represents a single biological replicate from individual mice. $N = 3$ mice for each sample. Unpaired *t* test was performed and resulting *P* values are as shown. ****$P < 0.0001$; ***$P < 0.001$; **$P < 0.01$; *$P < 0.1$; ns, not significant. Significant *P* values are shown in red. Error bars are SEM. *Cdkn1* 3 h vs 0 h. Spleen: p53 + /+, $P < 0.0001$****; p53m/+, $P = 0.0348$*; p53m/−, $P = 0.6060$(ns). Thymus: p53 + /+, $P = 0.0739$(ns); p53m/+, $P = 0.0228$*; p53m/−, $P = 0.3216$(ns). Bone Marrow: p53 + /+, $P = .0229$*; p53m/+, $P = 0.0220$*; p53m/−, $P = 0.4501$(ns); Mdm2/Mdm2, $P = 0.0373$*. *Bbc3* 3 h vs 0 h. Spleen: p53 + /+, $P = 0.0148$*; p53m/+, $P = 0.0693$*; p53m/−, $P = 0.3791$(ns). Thymus: p53 + /+, $P = 0.0953$(ns); p53m/+, $P = 0.0004$***; p53m/−, $P = 0.3346$(ns). Bone Marrow: p53 + /+, $P = 0.0279$*; p53m/+, $P = 0.0278$*; p53m/−, $P = 0.0003$***; Mdm2/Mdm2, $P = 0.0128$*. *Ccng1* 3 h vs 0 h. Spleen: p53 + /+, $P = 0.0015$**; p53m/+, $P = 0.0518$*; p53m/−, $P = 0.8721$(ns). Thymus: p53 + /+, $P = 0.1485$(ns); p53m/+, $P = 0.0211$*; p53m/−, $P = 0.4820$(ns). Bone Marrow: p53 + /+, $P = 0.0976$*; p53m/+, $P = 0.6494$(ns); p53m/−, $P < 0.0001$****; Mdm2/Mdm2, $P = 1336$(ns). (**D–F**) The corresponding RNA-seq results for *Cdkn1a* (**D**), *Bbc3* (**E**), and *Ccng1* (**F**) in bone marrow are shown. Statistical analysis for differential gene expression analysis was performed using DESeq2 (Love et al, 2014). (**G, H**) 8 week old mice of the indicated genotype were untreated or treated with 9.5 Gy of X-ray. After 3 h, RNA was extracted from bone marrow, and subjected to RT-qPCR. Expression of S*nora64* (**G**) and *Mroh8* (**H**) were determined relative to *Gapdh*. Each circle represents a single biological replicate from individual mice. Unpaired *t* test was performed and resulting *P* values are as shown. ****$P < 0.0001$; ***$P < 0.001$; **$P < 0.01$; *$P < 0.1$; ns, not significant. Significant *P* values are shown in red. Error bars are SEM. *Snora64* 3 h vs 0 h: p53 + /+, p-0.1889(ns); p53m/+, $P = 0.1889$(ns); p53m/−, $P = 0.0979$*. *Mroh8* 3 h vs 0 h: p53 + /+, $P = 0.5814$(ns); p53m/+, $P = 0.0099$**; p53m/−, $P = 0.6983$(ns). (**I**) 8 week-old mice of the indicated genotype were treated with 9.5 Gy of X-ray. After 3 h, bone marrows were subjected to cross-linking and ChIP-seq analyses. ChIP profiles of S*nora64* and *Mroh8* visualized in the IGV Browser are shown. Nrf1 occupancy is taken from an ENCODE data set for MEL cells (Accession: ENCSR135SWH). Data information: In (**A–C, G, H**), data are presented as mean ± SEM. *$P < 0.1$ (Student's *t* test).

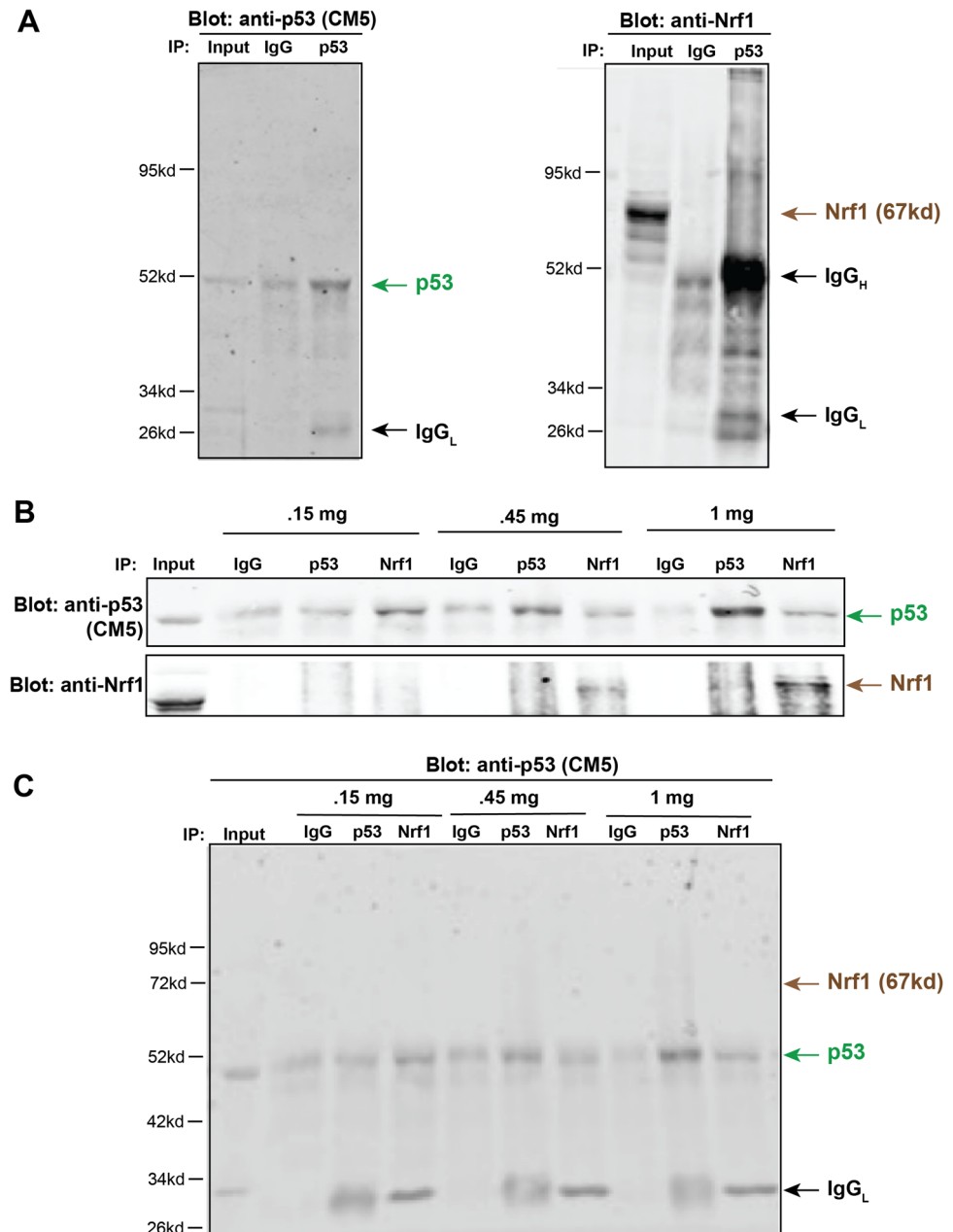

**Figure EV3. The antibody to p53 used for chromatin immunoprecipitation does not cross-react with Nrf1.**

(A) Protein extracts were prepared from the mouse metastatic esophageal squamous carcinoma cell line 378 and subjected to immunoprecipitation with CM5, an antibody to mouse p53 that was used in the chromatin immunoprecipitation assays or an IgG negative control. 10% of the extract that is used for immunoprecipitation is shown as Input. Immunoprecipitates were then subject to immunoblotting with either the antibody to p53 (CM5, on the left) or to Nrf1 (on the right). (B) The indicated amounts of protein extracts of 378 cells were subjected to immunoprecipitation with the shown antibodies. Immunoprecipitates were subjected to immunoblotting with either anti-p53 (CM5) or anti-Nrf1 antibodies as shown. (C) The immunoblot in (B) that was probed with the antibody to p53 prior to cropping is shown. The size of mouse Nrf1 (67kd) is indicated on the right. The position of standard molecular weight markers is shown on the left.

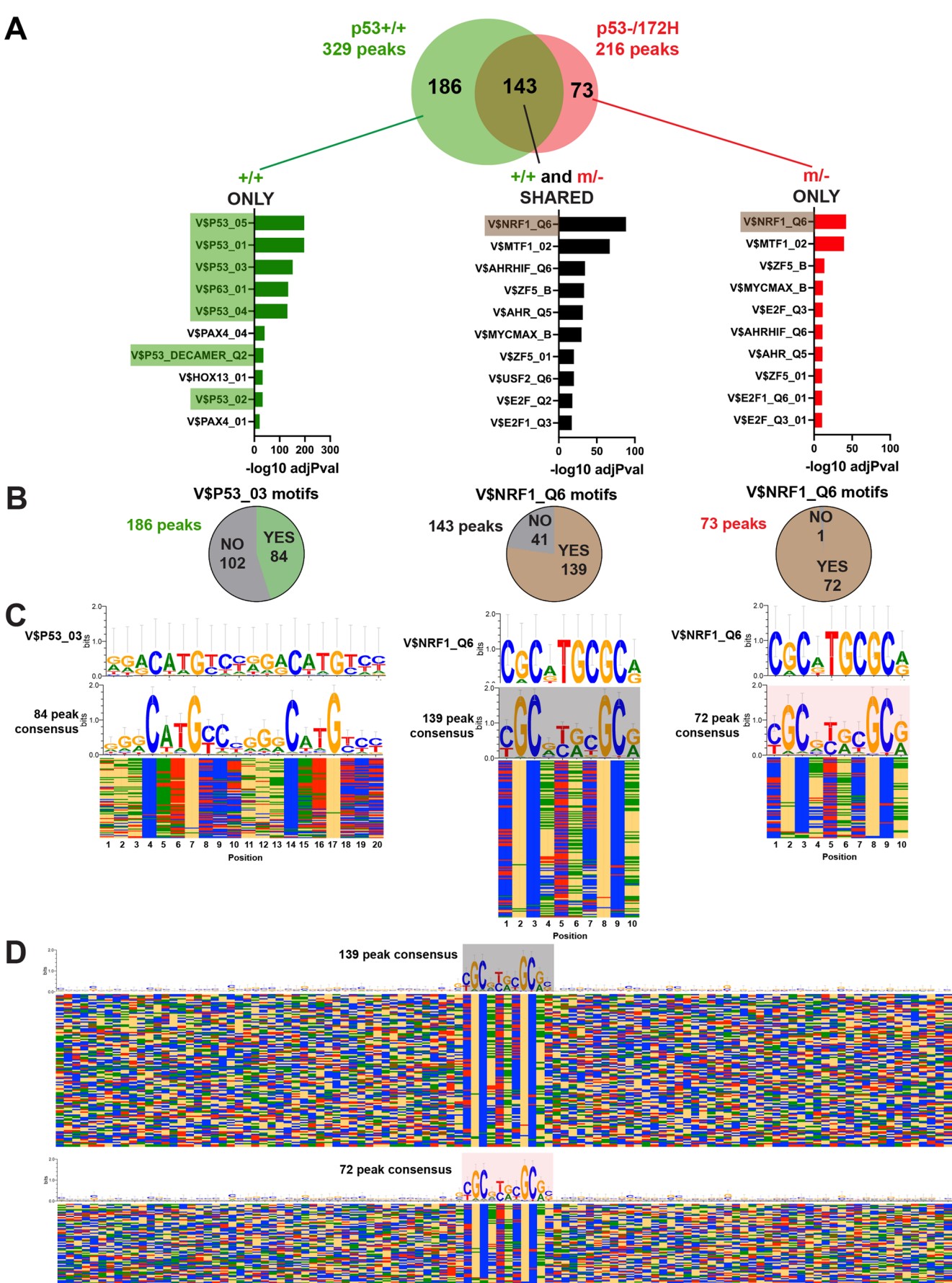

**Figure EV4.  Wild-type and mutant p53 show overlapping gene occupancies involving sequences that match an Nrf1 consensus motif.**

(A) 8 week-old mice of the indicated genotype were treated with 9.5 Gy of X-ray. After 3 h, bone marrows were subjected to cross-linking and ChIP-seq analyses. A Venn diagram is shown for genotype-specific and overlapping ChIP-seq peaks with an adjusted *P* value <0.2. The genomic sequences underneath the peaks detected by ChIP-seq were subjected to TRAP analysis. The top matrices that were detected with the corresponding adjPvalues are shown. Statistical analysis for motif analysis was done by TRAP (Transcription Factor Affinity Prediction) using web tools provided by the Max Planck Institute for Molecular Genetics (http://trap.molgen.mpg.de) (Thomas-Chollier et al, 2011). (B) As determined by TRAP analysis, the number of peaks which contain a consensus p53 motif (V$P53_03) or a consensus Nrf1 (V&NRF1_Q6) are shown as pie charts. (C) The sequences for the peaks with either a canonical p53 or Nrf1 motif, as indicated, were subjected to WebLogo3 analysis and the resulting consensus is shown. For comparison, the TRANSFAC consensus p53 motif (V$P53_03) or consensus Nrf1 (V&NRF1_Q6) are shown. Heat maps for the actual sequences are presented below the consensus. (D) An additional 50 bp surrounding the sequences for the peaks with Nrf1 motif were subjected to WebLogo3 analysis and the resulting consensus is shown. Heat maps for the actual sequences are presented below the consensus.

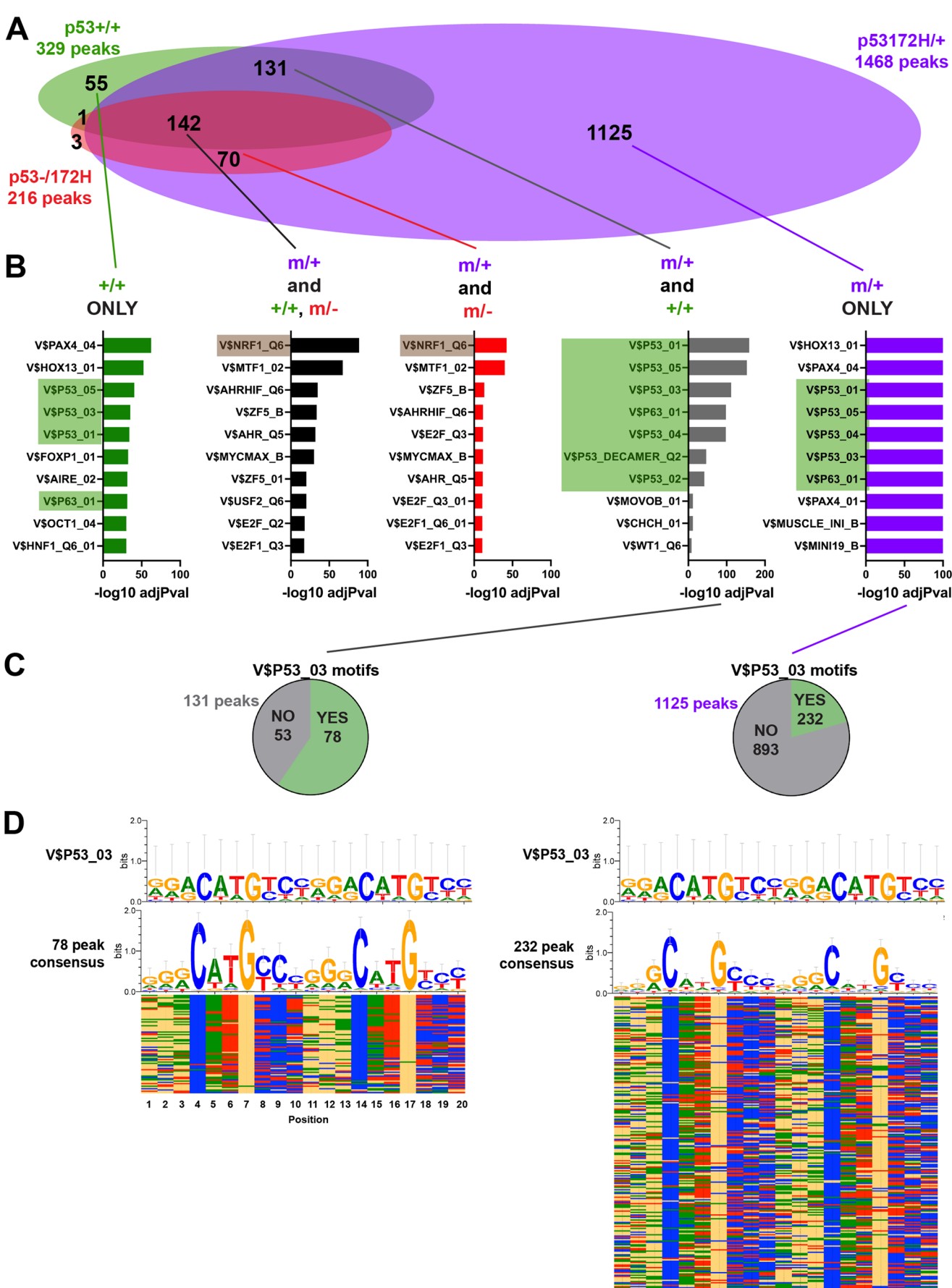

◀  **Figure EV5.   In the presence of wild-type p53, mutant p53 shows increased gene occupancies involving sequences that match a p53 consensus motif.**

(**A**) 8 week-old mice of the indicated genotype were treated with 9.5 Gy of X-ray. After 3 h, bone marrows were subjected to cross-linking and ChIP-seq analyses. A Venn diagram is shown for genotype-specific and overlapping ChIP-seq peaks with an adjusted *P* value <0.2. Statistical analysis for motif analysis was done by TRAP (Transcription Factor Affinity Prediction) using web tools provided by the Max Planck Institute for Molecular Genetics (http://trap.molgen.mpg.de) (Thomas-Chollier et al, 2011). (**B**) The genomic sequences underneath the peaks detected by ChIP-seq were subjected to TRAP analysis. The top matrices that were detected with the corresponding adjPvalues are shown. (**C**) As determined by TRAP analysis, the number of peaks which contain a consensus p53 motif (V$P53_03) are shown as pie charts. (**D**) The sequences for the peaks with a canonical p53 were subjected to WebLogo3 analysis and the resulting consensus is shown. For comparison, the TRANSFAC consensus p53 motif (V$P53_03) is shown. Heat maps for the actual sequences are presented below the consensus.

