## [Peer Review File · EMBO Reports]

Mutant p53 regulates a distinct gene set by a mode of genome occupancy that is shared with wild type

Ramy Rahmé, Lois Resnick-Silverman, Vincent Anguiano, Moray Campbell, Pierre Fenaux, and James Manfredi

Corresponding author(s): James Manfredi (james.manfredi@mssm.edu)

Review Timeline:

Submission Date:	28th Jul 23
Editorial Decision:	5th Sep 23
Revision Received:	14th Aug 24
Editorial Decision:	26th Nov 24
Revision Received:	5th Jan 25
Accepted:	14th Jan 25

Editor: Achim Breiling / Deniz Senyilmaz Tiebe

Transaction Report:

Dear Prof. Manfredi,

Thank you for the submission of your research manuscript to our journal, which was now seen by three referees, whose reports are copied below.

My colleague Achim is currently out of office. Therefore, I have stepped in as the secondary handling editor of your manuscript.

The referees express interest in the proposed differences in gene occupancy of mutant p53 or Mdm2 and wild type p53 and their regulation of gene expression. However, they also raise significant concerns that need to be addressed to consider publication here. In particular, all referees point out issues regarding missing controls and necessary validations.

Should you be able to address all criticisms in full, we would like to invite you to submit a revised manuscript. Please revise your manuscript with the understanding that the referee concerns (as in their reports) must be fully addressed and their suggestions taken on board. Please address all referee concerns in a complete point-by-point response. Acceptance of the manuscript will depend on a positive outcome of a second round of review. It is EMBO reports policy to allow a single round of major experimental revision only and acceptance or rejection of the manuscript will therefore depend on the completeness of your responses included in the next, final version of the manuscript.

We realize that it is difficult to revise to a specific deadline. In the interest of protecting the conceptual advance provided by the work, we recommend a revision within 3 months. Please discuss the revision progress ahead of this time with me if you require more time to complete the revisions, or if you have questions or comments regarding the revision (also by video chat).

1. A data availability section providing access to data deposited in public databases is missing (where applicable).
2. Your manuscript contains statistics and error bars based on $n=2$. Please use scatter plots in these cases.

You can submit the revision either as a Scientific Report or as a Research Article. For Scientific Reports, the revised manuscript can contain up to 5 main figures and 5 Expanded View figures, and it should not exceed 27000 characters. If the revision leads to a manuscript with more than 5 main figures it will be published as a Research Article. In this case the Results and Discussion section should be separate. If a Scientific Report is submitted, these sections have to be combined. This will help to shorten the manuscript text by eliminating some redundancy that is inevitable when discussing the same experiments twice. In either case, all materials and methods should be included in the main manuscript file.

4) a .docx formatted letter INCLUDING the reviewers' reports and your detailed point-by-point responses to their comments. As part of the EMBO publication's Transparent Editorial Process, EMBO reports publishes online a Review Process File (RPF) to accompany accepted manuscripts. This File will be published in conjunction with your paper and will include the referee reports, your point-by-point response and all pertinent correspondence relating to the manuscript. <https://www.embopress.org/page/journal/14693178/authorguide#transparentprocess>

5) a complete author checklist, which you can download from our author guidelines <https://www.embopress.org/page/journal/14693178/authorguide>. Please insert information in the checklist that is also reflected in the manuscript. The completed author checklist will also be part of the RPF.

6) Please note that all corresponding authors are required to supply an ORCID ID for their name upon submission of a revised manuscript (<<https://orcid.org/>>). Please find instructions on how to link your ORCID ID to your account in our manuscript tracking system in our Author guidelines <<https://www.embopress.org/page/journal/14693178/authorguide#authorshipguidelines>>

7) Before submitting your revision, primary datasets produced in this study need to be deposited in an appropriate public database (see <https://www.embopress.org/page/journal/14693178/authorguide#datadeposition>). Please remember to provide a reviewer password if the datasets are not yet public. The accession numbers and database should be listed in a formal "Data Availability" section placed after Materials & Method (see also <https://www.embopress.org/page/journal/14693178/authorguide#datadeposition>). Please note that the Data Availability Section is restricted to new primary data that are part of this study. * Note - All links should resolve to a page where the data can be accessed. *
If your study has not produced novel datasets, please mention this fact in the Data Availability Section.

Additional information on source data and instruction on how to label the files are available:
<https://www.embopress.org/page/journal/14693178/authorguide#sourcedata>

9) Our journal encourages inclusion of *data citations in the reference list* to directly cite datasets that were re-used and obtained from public databases. Data citations in the article text are distinct from normal bibliographical citations and should directly link to the database records from which the data can be accessed. In the main text, data citations are formatted as follows: "Data ref: Smith et al, 2001" or "Data ref: NCBI Sequence Read Archive PRJNA342805, 2017". In the Reference list, data citations must be labeled with "[DATASET]". A data reference must provide the database name, accession number/identifiers and a resolvable link to the landing page from which the data can be accessed at the end of the reference. Further instructions are available at <http://www.embopress.org/page/journal/14693178/authorguide#referencesformat>

- the name of the statistical test used to generate error bars and P values,
- the number (n) of independent experiments (please specify technical or biological replicates) underlying each data point,
- the nature of the bars and error bars (s.d., s.e.m.),
- If the data are obtained from n Program fragment delivered error `Can't locate object method "less" via package "than" (perhaps you forgot to load "than"?) at //ejpvfs23/sites23b/embor_www/letters/embor_decision_revise_and_review.txt line 56.' 2, use scatter blots showing the individual data points.

12) Please also note our reference format:

I look forward to seeing a revised version of your manuscript when it is ready. Please let me know if you have questions or comments regarding the revision.

Kind regards,

Deniz Senyilmaz Tiebe

Deniz Senyilmaz Tiebe, PhD
Scientific Editor
EMBO Reports

Referee #1:

In the manuscript 'Wildtype and tumor-derived mutant p53 share a non-canonical mode of gene occupancy' the authors use mouse models of WT p53, overexpression of MDM2 and mutant p53 to determine p53 mediated transcription in bone marrow after radiation stress. The RNA-seq and chip-seq profiles have been thoroughly analysed and compared to each other and reveal interesting novel regulation pathways as well as novel leads, especially within the Nrf1 pathway. The manuscript is well-written and the data looks conclusive. However, as the study stands, some controls and validations are missing.

Major comments:

1. The authors were not able to verify p53 status or more importantly p53 levels in the bone marrow of the mice. It is therefore hard to validate the importance of p53 and p53 activation in each of the mouse models. Do the authors have other tissue they could use as a surrogate and show activation of the different p53s in the different animal models?
2. In supplemental figure S3 the authors can see delta 10 in the thymus running at a lower level than WT. Are the bands seen in delta10 + irradiation actually delta 10 p53? They seem to be running lower than the thymus positive control. The text suggest no p53 was detected, but I am not certain that is true. Are these breakdown products/ other isoforms of p53? If other isoforms this is a bit troublesome as the effects seen might then be more dependent on other isoforms than on the C-terminal variants. This needs further clarification.
3. It would be good to validate some of the hypotheses generated from this work. Can the authors validate regulation of Nrf1 targets in the 172/- model and MDM2/MDM2 model using qRT PCR or protein analysis in an independent experiment.
4. Are the effects on gene expression confined the the bone marrow or are they observed in other tissues?

Minor comments:

1. Figure 5H is missing

Referee #2:

This is a well-presented set of data in which the authors attempted to shed light on gene occupancy within the interplay between wt-p53, mutantp53 and MDM2 in non-malignant cells. The manuscript combined CHIP- and RNA-seq analyses using murine bone marrows derived from isogenic in vivo models. Response to radiation has been proposed as biological readout of the proposed in vivo model system.

Listed below is a number of specific points that need to be addressed to strength the conclusions proposed by the authors.

1. CHIP-seq analyses need to be complemented with additional antibodies related to proteins depicting chromatin state and the existence of transcriptional competent complexes. This is specifically required to evaluate the expanded mut-p53 related program compared to that of wt-p53.
2. 172H represents human 175H which is the prototype of one of the main class of human tumour derived p53 mutants such as defective structure mutant p53 proteins. What about 273H or 248W, do they behave as 175H?
3. The authors pointed out that their findings could strongly contribute to clarify wt-p53, mutant p53 and MDM2 interplay in gene occupancy in non-malignant cells herein resumed by murine bone marrows. This statement needs to be further validated by using non-malignant human cells. If the related findings will confirm those reported by the authors, the overall scientific value of

the manuscript will be certainly improved.

4. The authors should evidence if there is any significant difference in the transcriptional program of 172H/+ versus 172H/- mouse bone marrows upon radiation treatment.
5. One limitation of the proposed experimental model resides on the lack of a different half-life between wt- and mutant p53 proteins as occurs in human cancers. As a matter of a diverse amount of two mentioned proteins this might have an important impact on the activation of specific transcriptional programs especially for those bindings onto non-canonical p53 binding sites.
6. MDM2 blunted the expression of only a subset of specific target genes. From what has been previously reported wt-p53 should be more prone to MDM2-mediated degradation than mut-p53. Is there any difference in the ability of MDM2 to abrogate transcriptional programs accordingly to the TP53 status of the mouse bone marrows?

Referee #3:

In the article entitled "WT and tumor-derived mutant p53 share a non-canonical mode of gene occupancy", Ramy Rahme et al., investigate the interplay between mutant p53 or MDM2 and wild-type p53 in gene occupancy and expression in response to radiation using isogenic mouse strains. The authors claim that contrary to a commonly accepted dominant -negative effect, mutant p53 and MDM2 enhances the occupancy of WT p53 on many canonical p53 target genes, while they inhibit only a subset of WT p53 target gene expression. The authors identified that the C-terminal 19 amino-acids of p53 suppress the p53 mediated survival to sublethal doses of radiation. The authors also claim that the mutant p53R172H, (homologous to the hot-spot p53 mutation in human cancer, p53 R175H) regulate expression of genes by binding to non-canonical p53 response element in their regulatory regions. The authors claim that the heterozygous p53 R172H/+ genotype has an expanded transcriptome compared to WT p53 genotype (p53+/+).

The data support some of the conclusions and part of the title. Revisions are required.

Fig1A:

- the MDM2/MDM2 mice are not described in the material and methods and in the figure legend.
- the sex of the mice is not indicated
- The Bonferroni's correction is not necessary in my view as the mouse model experiment explores only one question, i.e.the impact of p53 whether wt, mutant or null.

Suppl fig1. The immunoblot is not horizontal and is cut too close to the 53kD and the 40kD The authors should show the entire blot (20kD to 60kD).

The rabbit polyclonal CM5 antibody is well specific of mouse p53, detecting several epitopes in the N-terminus and C-terminus of mouse p53. CM5 antibody has a high affinity for mouse p53 proteins. CM5 detects all isoforms of mouse p53 (splice variants and post-translational modifications) therefore it is normal that several bands corresponding to p53 proteins are detected between 25kD and 53kD when using CM5. However, to convince readers that the authors used appropriately the CM5 antibody (appropriate buffer and antibody concentration) in the ChIP-seq experiments, the authors should show immunoprecipitation assay of mouse p53 protein with CM5 antibody. The authors could also perform westernblot from mouse cells transfected with p53siRNA that will clearly demonstrate the specificity of the CM5 antibody.

Fig2H: to convince readers that the CM5 antibody does not bind NRF1 protein, the authors should perform immunoprecipitation assay with NRF1 antibody and blot with CM5 antibody or NRF1 antibody. (mouse NRF1 protein is 534aa, will migrate above p53 proteins)

The NRF1 response element consensus sequence (CGCaTGCGC) is similar to half of the p53 response element consensus sequence, therefore it is not surprising that p53 protein could bind specifically to NRF1 response element particularly if successive repeat of NRF1 response element are present in the DNA sequence such as a repeat of two NRF1 aCGCATGCGCaCGCATGCGC fit also the p53 response element consensus sequence. The authors should publish 50bp upstream and 50 bp downstream of the DNA sequence containing the NRF1 binding site to determine whether such sequence could also be a p53 response element.

It is possible that the p53R172H could physically interact and form a protein complex with NRF1. The authors could investigate by co-immunoprecipitation assay whether p53R172H can bind with NRF1.

Fig4: Similarly, to convince reader that CM5 does immunoprecipitate p53D10 and p53D19, the authors should perform immunoprecipitation assay using CM5 and blot with other p53 polyclonal antibody (do not use 421 monoclonal antibody as its epitope is deleted in p53D19, while the 421 epitope is present in p53D10).

Icahn
School of
Medicine at
Mount
Sinai

James J. Manfredi, Ph.D.
Professor
Department of Oncological Sciences

One Gustave L. Levy Place
Box 1130
New York, NY 10029-6574

Phone 212.659.5495
Facsimile 212.987.2240
Email james.manfredi@mssm.edu

August 6, 2024

Deniz Senyilmaz Tiebe, Ph.D.
Scientific Editor
EMBO Reports

Dear Dr. Tiebe,

We are submitting online for your consideration the revised version of our manuscript entitled "Wild type and tumor-derived mutant p53 share a non-canonical mode of gene occupancy" by Ramy Rahmé, Lois Resnick-Silverman, Vincent Anguiano Moray J. Campbell, Pierre Fenaux, and myself. This manuscript represents original research, has not been previously published, and is not being considered for publication elsewhere. The authors declare no conflict of interest.

We apologize for the delay in revising the manuscript. But, as to be expected with exclusively in vivo studies, the breeding of mice with the appropriate genotypes involved considerable time and expense.

This is a summary of the revisions to the manuscript:

1. **ADDED AUTHOR:** We have added an additional author, Lois Resnick-Silverman, who was instrumental in generating the additional new data to address reviewers' concerns.
2. **NEW FIGURES:** The new data is contained in five Extended View Figures that were not part of the original submission.
3. **METHODS:** Additions have also been made to the Methods section that cover the experimental approaches that were not included in the first version. These are relevant to the new Extended View Figures.
4. **EXISTING FIGURES:** Figure 5 as well as Appendix Figures S1 and S3 have been modified in response to reviewers' comments.
5. **RESULTS and DISCUSSION:** Additions and edits were made in response to reviewers' concerns.

Please see Point-by-point responses to reviewers' comments below.

The p53 tumor suppressor is commonly mutated in human cancer. The persistent expression of a missense mutant p53 in tumors has led to the notion that such genetic alterations may confer gain-of-function oncogenic activity in addition to mere loss-of-function. A growing body of evidence now

supports a role for mutant p53 in occupying genes and regulating their transcription. This is although these mutants lack intrinsic DNA binding activity. However, the mechanism by which tumor-derived mutant p53 exerts its transcriptional effects has been elusive. Previous studies have relied largely on the use of cancer cell lines in culture. This has resulted in considerable confusion in the field, as different laboratories proposing distinct and sometimes unrelated mechanisms for mutant p53 transcriptional activity.

Here we attempt to provide some clarity to this significant issue by studying the radiation response in vivo using mouse strains that are otherwise identical except for their p53 status. This has allowed some clarity to be brought to a complex problem. We have performed a global transcriptomic and cistromic analysis of p53 in the bone marrow. This was performed in the context of sensitivity to a lethal dose of whole-body radiation. This approach proved challenging, as it was the first time, that we are aware, that such analyses of p53 have been performed in vivo. More importantly, this allowed a direct comparison between wild type and mutant p53 in gene occupancy and regulation.

This has led to key findings:

- Unexpectedly, mutant p53 and the negative regulator Mdm2 only inhibit a subset of wild type p53 gene expression.
- The presence of either mutant p53 or Mdm2 enhances the occupancy of wild type p53 to canonical targets.
- Mutant p53 indeed occupies genes and regulates their expression but does so via a non-canonical means that is remarkably shared between wild type and mutant p53.
- The heterozygous 172H/+ genotype has an expanded transcriptome compared to wild type p53+/+. This is due to a retention of canonical gene regulation but with the addition of non-canonical transcriptional effects associated with the mutant p53.

At a journal such as EMBO Reports, one expects authors often try to emphasize and sometimes exaggerate the novelty of their studies. In our case, we would like to point out that this is the first study to our knowledge that performs an integrated RNA-seq and ChIP-seq analysis on p53 using (1) non-cancerous bone marrow with (2) tissues obtained in vivo rather than cell lines, and (3) allows for a direct comparison between wild type and mutant p53 functions. This has been done in the biologically relevant setting of radiation sensitivity.

Coupled with the significance and translational potential in studying the radiation response in vivo and the importance of understanding the functional consequences of missense mutation of p53 in human cancer, we feel that our revised study is now well suited for publication in EMBO Reports. We hope you agree.

Sincerely,

Point-by-Point Response to Previous Reviews

Referee #1:

In the manuscript 'Wildtype and tumor-derived mutant p53 share a non-canonical mode of gene occupancy' the authors use mouse models of WT p53, overexpression of MDM2 and mutant p53 to determine p53 mediated transcription in bone marrow after radiation stress. The RNA-seq and chip-seq profiles have been thoroughly analyzed and compared to each other and reveal interesting novel

regulation pathways as well as novel leads, especially within the Nrf1 pathway. The manuscript is well-written, and the data looks conclusive. However, as the study stands, some controls and validations are missing.

Major comments:

1. The authors were not able to verify p53 status or more importantly p53 levels in the bone marrow of the mice. It is therefore hard to validate the importance of p53 and p53 activation in each of the mouse models. Do the authors have other tissue they could use as a surrogate and show activation of the different p53s in the different animal models?

We share the reviewer's concern about the inability to detect p53 protein expression, wild-type or mutant, in bone marrow samples (**Appendix Figure S1**). As suggested by the reviewer, we have examined spleen and thymus from mice irradiated under the same conditions as the bone marrow studies. Both immunoblotting for protein (**Extended Figure 1**) and RT-qPCR for mRNA expression (**Extended Figure 2**) were performed. Wild-type and mutant p53 protein expression was detectable and is similar in both tissues, both before and after treatment (**Extended Figure 1A**). Overexpression of Mdm2 in the transgenic mouse model appears to modestly affect p53 levels in spleen and thymus under these same conditions (**Extended Figure 1C**).

RT-qPCR analysis of bone marrow confirms findings with the various genotypes (p53^{+/+}; p53^{m/+}; p53^{m/-}; and Mdm2 Tg/Tg) RNA-seq for three targets (*Cdkn1a*, *Bbc3*, and *Ccng1*) (**Extended Figure 2A-F**). Spleen and thymus with either the wild-type or the heterozygous m/+ genotypes show robust activation of mRNA after irradiation (**Extended Figure 2A, C, E**) and this is reflected in protein increases for one target, p21 (*Cdkn1a*) (**Extended Figure 1A**). As expected, tissues from the m/- mice show no mRNA upregulation (**Extended Figure 2A, C, E**), nor increases in p21 protein (**Extended Figure 1A**). In the Mdm2/Mdm2 tissues, upregulation of p21 protein is impaired, however modest increases in Puma are retained (**Extended View Figure 1C**).

These results are now included in the text. Taken together, these new data provide two additional insights. First, they confirm that effects of mutant p53 are not due to differences in protein expression. Rather, observed effects are likely due to differences in p53 activity. Second, they show that mutant p53 protein is induced to comparable levels as wild-type regardless of genotype of tissues (m/+, m/-).

2. In supplemental figure S3 the authors can see delta 10 in the thymus running at a lower level than WT. Are the bands seen in delta10 + irradiation actually delta 10 p53? They seem to be running lower than the thymus positive control. The text suggests no p53 was detected, but I am not certain that is true. Are these breakdown products/ other isoforms of p53? If other isoforms this is a bit troublesome as the effects seen might then be more dependent on other isoforms than on the C-terminal variants. This needs further clarification.

The reviewer is correct in their interpretation of this figure (**Appendix Figure S3**), and we have modified the figure itself as well as its discussion in the text. As is clear from the immunoblot, protein expression of both the $\Delta 10$ and $\Delta 19$ engineered forms of wild-type p53 can be detected. Although there is some variation from mouse to mouse, it appears that both forms are upregulated in response to radiation and their levels of expression are substantially higher than that of wild-type p53 (**Appendix Figure S3**). This altered expression is a likely explanation for the radiation-related phenotypes of the mice expressing the truncated proteins (**Figure 4A-B, F**). Nevertheless, the RNA-seq studies identify genes that are selectively upregulated in p53^{+/+} bone marrow (**Figure 4C-D**). This suggests that there may also be intrinsic differences in p53 activity between the wild-type and $\Delta 19$ proteins. Additions have been made to the text to better emphasize the role of increased protein expression in the observed effects of $\Delta 10$ and $\Delta 19$.

3. It would be good to validate some of the hypotheses generated from this work. Can the authors validate regulation of Nrf1 targets in the 172/- model and MDM2/MDM2 model using qRT PCR or protein analysis in an independent experiment.

This is an important point. In response, we have performed RT-qPCR analysis to validate effects seen with RNA-seq. In the bone marrow, RT-qPCR for each genotype shows comparable effects as what is seen by RNA-seq for three targets (*Cdkn1a*, *Bbc3*, and *Ccng1*) (**Extended View Figure 2A-F**). We performed RT-qPCR experiments to show that there are targets that are only upregulated in the m/- bone marrow (*Snora64*, **Extended View Figure 2G**), but not in +/+ or m/+ tissue. Conversely, there is a target *Mroh8* that is only upregulated in the m/+ bone marrow but not in either +/+ or m/- tissue (**Extended View Figure 2H**). This confirms the existence of a mutant p53-specific target (*Snora64*) as well as a target that is only seen in the heterozygous m/+ bone marrow, *Mroh8*. To specifically address the reviewer's concern, both these targets show an overlap between the p53 ChIP peak and that of Nrf1 (**Extended View Figure 2I**). Additions to the text have been made to discuss this new data.

4. Are the effects on gene expression confined to the bone marrow or are they observed in other tissues?

Please see discussion under Item 1 above. In short, RT-qPCR confirms similar effects in spleen and thymus as in bone marrow (**Extended View Figure 2A-F**).

Minor comments:

1. Figure 5H is missing

The original **Figure 5** was mis-labeled. We apologize for this. A revised **Figure 5** is now included.

Referee #2:

This is a well-presented set of data in which the authors attempted to shed light on gene occupancy within the interplay between wt-p53, mutant p53 and MDM2 in non-malignant cells. The manuscript combined CHIP- and RNA-seq analyses using murine bone marrows derived from isogenic in vivo models. Response to radiation has been proposed as biological readout of the proposed in vivo model system.

Listed below is a number of specific points that need to be addressed to strength the conclusions proposed by the authors.

1. CHIP-seq analyses need to be complemented with additional antibodies related to proteins depicting chromatin state and the existence of transcriptional competent complexes. This is specifically required to evaluate the expanded mut-p53 related program compared to that of wt-p53.

Elucidating underlying molecular mechanisms for the ability of mutant p53 to execute a transcriptional program and how this compares to that of wild-type is clearly the long term goal that is justified by the studies presented in our manuscript. It was necessary to establish first that mutant p53 does exert transcriptional effects that are associated with occupancy of genomic sites. This is something we hope the reviewers agree we have accomplished with the considerable data provided in the manuscript. We would like to argue that such ChIP-seq studies as proposed by the reviewer are indeed essential to understand the molecular basis of the effects seen here. The considerable sequencing expense, bioinformatics effort, and mouse costs make this avenue of study a logical next step rather than necessary for the current study. We apologize for emphasizing the costs of these efforts. We wholeheartedly agree they are necessary and should be pursued. We just hope the reviewer agrees that they probably are outside the scope of the conclusions of the current study.

2. 172H represents human 175H which is the prototype of one of the main class of human tumour derived p53 mutants such as defective structure mutant p53 proteins. What about 273H or 248W, do they behave as 175H?

The reviewer raises an important point. There is a growing body of evidence that not all tumor-derived mutant p53 proteins have similar activities. The translational potential of findings

with mutant p53 thus will rely on teasing out general versus mutant-specific effects. That said, it can be argued that such a question is beyond the scope of the current study. To explore findings with other mutants would involve extensive additional breeding, not to mention, cost, and is probably best left for another day. However, the notion of whether our findings can be generalizable or are specific to 172H does need to be discussed. Accordingly, we have added text to the Discussion addressing this. It should also be noted that this one mutation, 175H in humans, probably represents approximately 400,000 patients a year. Thus, elucidating its functions, in and of itself, can still be considered a worthwhile endeavor.

3. The authors pointed out that their findings could strongly contribute to clarify wt-p53, mutant p53 and MDM2 interplay in gene occupancy in non-malignant cells herein resumed by murine bone marrows. This statement needs to be further validated by using non-malignant human cells. If the related findings will confirm those reported by the authors, the overall scientific value of the manuscript will be certainly improved.

Determining the relevance of our studies in humans is, indeed, clearly the next and critical step. The challenge, of course, is the findings herein have particular biological relevance as these were performed in vivo using a genetically defined system. To replicate such studies in humans would thus be quite difficult. That said, the reviewer rightly proposes the use of non-malignant human cells. This is an important long-term goal that will require us to find suitable collaborators than can provide use with hematopoietic or bone marrow cells of human origin that are non-malignant. The added challenge, of course, is that they be isogenically matched as well. We have added some lines to the Discussion to emphasize the relevance of the mouse versus human distinction.

4. The authors should evidence if there is any significant difference in the transcriptional program of 172H/+ versus 172H/- mouse bone marrows upon radiation treatment.

This is an intriguing and important question. If one compares the genes that are occupied and regulated in the 172H/- bone marrow (**Figure 5A**) to that of 172H/+ bone marrow (**Figure 6A**), there are only two genes that overlap: *Nr1d2* and *Zcwpw1*. The data showing this shared regulation and occupancy for *Zbtb41* is shown in **Figure 6F**.

Further, of the 80 such occupied and upregulated genes in 172H/+, only 16 of these are also upregulated in the 172H/- bone marrow (the latter not necessarily being occupied by p53) (**Figure 6C, E**). It is unclear why we do not detect occupancy in the 172H/- mice for these 16 shared genes. The studies in bone marrow have been challenging and this may reflect the limitations of the ChIP-seq experiments. We have made additions to the text to discuss these findings in more detail. The data is consistent with the idea that there are transcriptional programs that are shared between 172H/+ and 172H/- as well as gene sets that are unique to each genotype. We thank the reviewer for drawing attention to this.

5. One limitation of the proposed experimental model resides on the lack of a different half-life between wt- and mutant p53 proteins as occurs in human cancers. As a matter of a diverse amount of two mentioned proteins this might have an important impact on the activation of specific transcriptional programs especially for those bindings onto non-canonical p53 binding sites.

This reviewer makes an extremely important point that was shared with Reviewer 1. To address this, albeit in a limited manner, we have now included immunoblotting for wild-type and mutant p53 in other tissues. Please see the response to Item 1 for Reviewer 1 above. In short, these new data show that levels of protein expression for p53 in either the +/+, 172H/-, or 172H/+ tissues is comparable, both in the untreated state and after irradiation.

6. MDM2 blunted the expression of only a subset of specific target genes. From what has been previously reported wt-p53 should be more prone to MDM2-mediated degradation than mut-p53. Is there any

difference in the ability of MDM2 to abrogate transcriptional programs accordingly to the TP53 status of the mouse bone marrows?

There is a growing body of evidence in the literature for p53-independent effects of Mdm2. A role for Mdm2 in regulating mutant p53 remains largely unexplored. Our experimental system is ideal to being to address this interesting question. One can consider this to be a logical next step requiring more extensive mouse breeding involving additional cost and time, rather than something germane to the current study. Nevertheless, we have added lines to the Discussion speculating about the possible effects of Mdm2 on mutant p53.

Referee #3:

In the article entitled "WT and tumor-derived mutant p53 share a non-canonical mode of gene occupancy", Ramy Rahme et al., investigate the interplay between mutant p53 or MDM2 and wild-type p53 in gene occupancy and expression in response to radiation using isogenic mouse strains. The authors claim that contrary to a commonly accepted dominant -negative effect, mutant p53 and MDM2 enhances the occupancy of WT p53 on many canonical p53 target genes, while they inhibit only a subset of WT p53 target gene expression. The authors identified that the C-terminal 19 amino-acids of p53 suppress the p53 mediated survival to sublethal doses of radiation. The authors also claim that the mutant p53R172H, (homologous to the hot-spot p53 mutation in human cancer, p53 R175H) regulate expression of genes by binding to non-canonical p53 response element in their regulatory regions. The authors claim that the heterozygous p53 R172H/+ genotype has an expanded transcriptome compared to WT p53 genotype (p53+/+).

The data support some of the conclusions and part of the title. Revisions are required.

Fig1A:

-the MDM2/MDM2 mice are not described in the material and methods and in the figure legend.

We apologize for this oversight. The mice are now discussed in detail in the Methods section with addition information provided in the figure legends.

- the sex of the mice is not indicated

This should have been included in the original submission. For ease of breeding and to avoid any confounding issues related to gender, all mice used in this study were male. This is now noted in the Methods section.

- The Bonferroni's correction is not necessary in my view as the mouse model experiment explores only one question, i.e.the impact of p53 whether wt, mutant or null.

The point is well-taken and the Bonferroni correction is no longer used.

Suppl fig1. The immunoblot is not horizontal and is cut too close to the 53kD and the 40kD The authors should show the entire blot (20kD to 60kD).

The blot has been re-cropped as requested.

The rabbit polyclonal CM5 antibody is well specific of mouse p53, detecting several epitopes in the N-terminus and C-terminus of mouse p53. CM5 antibody has a high affinity for mouse p53 proteins. CM5 detects all isoforms of mouse p53 (splice variants and post-translational modifications) therefore it is normal that several bands corresponding to p53 proteins are detected between 25kD and 53kD when using CM5. However, to convince readers that the authors used appropriately the CM5 antibody (appropriate buffer and antibody concentration) in the ChIP-seq experiments, the authors should show immunoprecipitation assay of mouse p53 protein with CM5 antibody. The authors could also perform westernblot from mouse cells transfected with p53siRNA that will clearly demonstrate the specificity of the CM5 antibody.

The reviewer makes an important point concerning the specificity of the CM5 antibody. To address this issue, we have included three additional pieces of data. First, an immunoblot using

CM5 that shows the full gel without cropping was performed to show that the only detectable band is that corresponding to the molecular weight of p53 (both wild-type and mutant). This is shown in the new **Extended View Figure 1B**. Second, use of tissues from p53-null mice shows that this band is no longer seen (**Extended View Figure 1C**). Third, extracts of mouse cells expressing mutant p53 were immunoprecipitated with CM5 and then these were subsequently immunoblotted with CM5. The only detectable band corresponds to the size of p53 (**Extended View Figure 3A, C**). Further, use of increasing amounts of extract, show a corresponding increase in the intensity of this band (**Extended View Figure 3B-C**). Taken together, these additional data argue in favor of CM5 being specific for full-length p53. This new information is now included in the text.

Fig2H: to convince readers that the CM5 antibody does not bind NRF1 protein, the authors should perform immunoprecipitation assay with NRF1 antibody and blot with CM5 antibody or NRF1 antibody. (mouse NRF1 protein is 534aa, will migrate above p53 proteins)

This is a critically important control. To address this, we have included two additional data. First, we note that there are no detectable bands with the migration size of Nrf1 (67kd) in immunoblots of tissues of various genotypes that have been probed with CM5 (**Extended View Figure 1B**). Second, an antibody to Nrf1 was used. Immunoprecipitation of extracts of mouse cells expressing mutant p53 with anti-Nrf1 and then subsequently probing with Nrf1 results in detection of a band of the appropriate molecular weight. Increasing amount of extract shows a corresponding increase in the intensity of this band, arguing that the antibody is detected Nrf1 (**Extended View Figure 3B**). When such Nrf1 immunoprecipitates are probed with CM5, no bands of the size corresponding to Nrf1 are detected (**Extended View Figure 3A-C**). This confirms that CM5 is probably not cross-reacting with Nrf1.

The NRF1 response element consensus sequence (CGCaTGCGC) is similar to half of the p53 response element consensus sequence, therefore it is not surprising that p53 protein could bind specifically to NRF1 response element particularly if successive repeat of NRF1 response element are present in the DNA sequence such as a repeat of two NRF1 aCGCATGCGCaCGCATGCGC fit also the p53 response element consensus sequence. The authors should publish 50bp upstream and 50 bp downstream of the DNA sequence containing the NRF1 binding site to determine whether such sequence could also be a p53 response element.

We thank the reviewer for drawing attention to the similarities of these consensus sites (**Appendix Figure S4**). To address this intriguing possibility, further analyses were performed (**Extended View Figures 4-5**). For this, all ChIP peaks were examined, regardless of association with regulated genes (Figure 1E). The genomic occupancies of p53^{+/+} and p53^{-/172H} were compared. A Venn diagram was generated to show specific and overlapping peaks for each genotype (**Extended View Figure 4A**). TRAP analysis for each set of peaks revealed that the ^{+/+} only peaks were enriched for p53 motifs, whereas the shared and m^{-/} peaks were high for Nrf1 motifs (**Extended View Figure 4A-B**). Of the 186 ^{+/+} peaks, 84 contained a match to the p53 consensus (**Extended View Figure 4C**). For the shared (143 peaks) and the m^{-/} only (73 peaks), the majority had sequences matching an NRF1 consensus (139, and 72 respectively) (**Extended View Figure 4C**). We then examined the surrounding 50bp as requested by the reviewer. There did not appear to be any further detectable consensus sequences beyond the 10bp NRF1 motif (**Extended View Figure 4D**). These data further support the central finding of this study, that wild-type and mutant p53 share a non-canonical means for genomic occupancy.

To validate the ability of such analyses to detect bona fide p53 motifs, peaks that contain p53 motifs that are shared between m^{+/} and ^{+/+} (78 out of 131 peaks) versus m^{+/} alone (232 out of 1125 peaks) were examined (**Extended View Figure 5**). The resulting consensus sequences match well to known p53 motifs (compare the 84 peak consensus in **Extended View Figure 4C** with that of both sets of peaks in **Extended View Figure 5C**). This contrasts with the consensus matches to the Nrf1 motif (**Extended View Figure 4C**). Although most of the shared peaks has a p53 motif,

intriguingly only a subset of the m/+ only peaks did. This suggest that perhaps the m/+ genotype results in yet another non-canonical mode for p53 genomic occupancy. These data also provide additional support for the idea that the m/+ genotype results in an expanded genomic occupancy as compared to that of the +/+genotype.

It is possible that the p53R172H could physically interact and form a protein complex with NRF1. The authors could investigate by co-immunoprecipitation assay whether p53R172H can bind with NRF1.

The possibility of a physical interaction between mutant p53 and Nrf1 would provide a simple explanation for the findings. To address this, immunoprecipitations of extracts from cells expressing mutant p53 were examined. Immunoprecipitation with anti-p53 antibody (CM5) and subsequent immunoblotting with anti-Nrf1, did not detect a band with the migration size corresponding to Nrf1 (**Extended View Figure 3B**). Immunoprecipitation with anti-Nrf1 antibody and subsequent immunoblotting with anti-53 (CM5) was less conclusive. Although a band with the molecular weight of p53 is detected with CM5 in these Nrf1 immunoprecipitates, it is unclear whether the intensity is above background. Further, the band intensity does not increase with the use of more extract. Taken together, there is little evidence to support the interaction between mutant p53 and Nrf1 at this time.

Fig4: Similarly, to convince reader that CM5 does immunoprecipitate p53D10 and p53D19, the authors should perform immunoprecipitation assay using CM5 and blot with other p53 polyclonal antibody (do not use 421 monoclonal antibody as its epitope is deleted in p53D19, while the 421 epitope is present in p53D10).

We apologize for any confusion in the discussion of the results with the Δ 10 and Δ 19 mice. Although eager to do so in the future, we have not yet done ChIP-seq studies with these genotypes. Hence, the requested evidence that CM5 efficiently immunoprecipitates either truncated protein has not been included in the revised manuscript.

Dear Prof. Manfredi,

Thank you for the submission of your revised manuscript to our editorial offices. I have now received the reports from the three referees that I asked to re-evaluate the study, you will find below. As you will see, referees #1 and #3 now support publication of the study in EMBO reports. In contrast, referee #2 states that the revised manuscript addressed only partially his/her comments, without being more specific. Going through your p-b-p-response, I consider the comments however as adequately addressed, i.e. I feel that the explanations why certain experiments were not performed are reasonable. I will therefore, also considering the positive evaluation of the other two referees, proceed with the manuscript.

Referee #3 has a final suggestion to improve the manuscript I ask you to address in a final revised manuscript.

Moreover, I have these editorial requests

- Please reduce the number of keywords to five.

- Please order the sections like this using these names:

Title page - Abstract - Keywords - Introduction - Results - Discussion - Methods - Data availability section (DAS) - Acknowledgements (including funding information) - Disclosure and Competing Interests Statement - References - Figure legends - Expanded View Figure legends

- We now use CRediT to specify the contributions of each author in the journal submission system. CRediT replaces the author contribution section. Please use the free text box to provide more detailed descriptions and do NOT provide your final manuscript text file with an author contributions section. See also our guide to authors:

<https://www.embopress.org/page/journal/14693178/authorguide#authorshippinguidelines>

- Please remove the highlights section from the manuscript. This can be re-used for the bullet points (see below).

- Per journal policy, we do not allow 'data not shown', which is stated twice in the manuscript. All data referred to in the paper should be displayed in the main or Expanded View figures, or an Appendix. Thus, please add these data (or change the text accordingly if these data are not central to the study). See:

<https://www.embopress.org/page/journal/14693178/authorguide#unpublisheddata>

- Please make sure that all figure panels (main, EV and Appendix figures) are called out separately and sequentially. Please use the expression 'Figure EVx' for the callouts of the EV figures (not 'Extended Figure x').

- There is a legend for Fig. EV6, but no such figure is uploaded. There are also no callouts to such a figure and its panels. Please check.

- There are callouts for Suppl. Fig. 5. Please update.

- Please add scale bars of similar style and thickness to all microscopic images (presently shown only in the Appendix, it seems), using clearly visible black or white bars (depending on the background). Please place these in the lower right corner of the images themselves. Please do not write on or near the bars in the image but define the size in the respective figure legend.

- Please check that the number "n" for how many independent experiments were performed, their nature (biological versus technical replicates), the bars and error bars (e.g. SEM, SD) and the test used to calculate p-values is indicated in the respective figure legends (main, EV and Appendix figures). Please also check that all the p-values are explained in the legend, and that these fit to those shown in the figure. Please provide statistical testing where applicable. Please avoid the phrase 'independent experiment', but clearly state if these were biological or technical replicates. Please also indicate (e.g. with n.s.) if testing was performed, but the differences are not significant. In case n=2, please show the data as separate datapoints without error bars and statistics. See also:

<http://www.embopress.org/page/journal/14693178/authorguide#statisticalanalysis>

If n<5, please show single datapoints for diagrams. Moreover:

- Please note that the legends for figures EV 2b-h, i are not provided in the sequential manner (legend for figure EV 2c, e, g-h, i is provided before legend of figure EV 2b, 2d, 2f; 2b, d, f respectively). This needs to be rectified.

- Please note that the figure EV 1i is mislabeled as figure EV 1a in the manuscript. This needs to be rectified.

- Please note that the exact p values are not provided in the legends of figures 1b; 4f; EV 2a, c, d.

- Please indicate the statistical test used for data analysis in the legends of figures 1c-e; 2a-c; 3a-b, e; 4e; 5a-c, h, j; 6a-b, d-f; 7a-d; EV 2b, d, f; EV 4a; EV 5b.

- Please note that for the figures 1a-b; 4a, f; p-values and statistical tests are indicated in the legends. However, comparison for the same, "****/**/****" has not been represented in the figures. Please rectify this in the figures or legends as applicable.

- Please note that information related to n is missing in the legends of figures 2a; 3a-b; 4e-f; 5h; 6d-e; 7d; EV 2b, d, f.
- Please note that the error bars are not defined in the legends of figures 3e; EV 2a, c, e, g-h.

- Please add to each legend (main, EV figures, where applicable) a 'Data Information' section explaining the statistics used or providing information regarding replicates and scales. See:

- All materials and methods used need to be described in the main text using our 'Structured Methods' format, which is required for all research articles. According to this format, the Methods section should include a Reagents and Tools Table (listing key reagents, experimental models, software, and relevant equipment and including their sources and relevant identifiers), uploaded as separate file, followed by a Methods section in which we encourage the authors to describe their methods using a step-by-step protocol format with bullet points, to facilitate the adoption of the methodologies across labs. More information on how to adhere to this format as well as downloadable templates (.doc or .xls) for the Reagents and Tools Table can be found in our author guidelines (section 'Structured Methods'):

- Please move the primer information (Table S1?) from the methods section to the reagents and tools table and add a callout (see 'reagents & tools table').

- Please use our reference format (using et al for more than 10 authors):

- Please remove the author names and affiliations from the Appendix title page. It is sufficient to mention the title of the paper. Moreover, please add page numbers to the Appendix file and to the table of contents. It would also be more comprehensible if figures and legends were shown on the same page. Please do this where possible.

- Please add direct links for the GEO datasets GSE281318 and GSE281319 and make sure these are public latest on the date of online publication of the manuscript.

In addition, I would need from you uploaded separately:

Best,

Referee #1:

The authors have carefully addressed all concerns and in the process uncovered exciting new conclusions that strengthen the manuscript.

Referee #2:

Minimal experimental work has been performed to address the specific comments raised by this reviewer. The authors have raised a number of issues justifying why they could not address the specific comments. Some of them are reasonable, but some left with the need of additional experimental efforts that could have been done. Overall, the revision addressed only partially the comments of this reviewer.

Referee #3:

The article is very interesting, and the revised manuscript shows significant improvement. The data strongly support the conclusions. The authors have addressed all of my comments thoroughly. However, I regret that the authors do not clearly emphasize in the title and abstract that 172H⁻ induces a distinct gene expression program compared to p53^{-/-} (p53 null). This finding unequivocally demonstrates that a conformational missense mutation of p53 exhibits different activities from a homozygous loss of the p53 gene, which is far less frequent in cancer compared to p53 missense mutation.

All editorial and formatting issues were resolved by the authors.

Prof. James Manfredi
Icahn School of Medicine at Mount Sinai
Oncological Sciences
Box 1130
One Gustave L. Levy Place
New York, NY 10029
United States

Dear Prof. Manfredi,

I am very pleased to accept your manuscript for publication in the next available issue of EMBO reports. Thank you for your contribution to our journal.

Yours sincerely,
